# Nationwide Screening for Arthropod, Fungal, and Bacterial Pests and Pathogens of Honey Bees: Utilizing Environmental DNA from Honey Samples in Australia

**DOI:** 10.3390/insects16080764

**Published:** 2025-07-25

**Authors:** Gopika Bhasi, Gemma Zerna, Travis Beddoe

**Affiliations:** 1Department of Ecological, Plant and Animal Science, School of Agriculture, Biomedicine and Environment, La Trobe University, Bundoora 3083, Australia; 20522812@students.latrobe.edu.au (G.B.); g.zerna@latrobe.edu.au (G.Z.); 2La Trobe Institute for Sustainable Agriculture and Food, La Trobe University, Bundoora 3083, Australia

**Keywords:** *Apis mellifera*, eDNA, honey, *Paenibacillus larvae*, *Melissococcus plutonius*, *Nosema apis*, *Nosema ceranae*, *Ascosphaera apis*, *Aethina tumida*, *Galleria mellonella*, health, surveillance

## Abstract

European honey bees are vital to Australian agriculture for honey production and crop pollination, yet little is known about their pathogens and pests. Analysing environmental DNA from 135 honey samples nationwide, this study found *Nosema ceranae* (57%) to be the most common pathogen, followed by pests *Aethina tumida* (40%) and *Galleria mellonella* (37%), and pathogens *Paenibacillus larvae* (21%), *Nosema apis* (19%), and *Melissococcus plutonius* (18%). *Ascosphaera apis* was rare (5%). The results outline the geographic distribution of key bee threats in Australia.

## 1. Introduction

In Australia, the agricultural sector heavily relies on insect pollination, particularly from the western honey bee *Apis mellifera* [1], which is essential for pollinating a diverse array of crops and enhancing agricultural productivity and food security nationwide [2]. The benefits of crop pollination extend beyond agriculture, exerting far-reaching impacts on the broader Australian community by facilitating pollination and positively influencing crop outcomes through their synergistic relationship with flowering plants [3]. Previous estimations indicate that the total value of paid and unpaid pollination services amounts to approximately AUD 1.2 billion annually in Australia [4]. The reliance on honey bees for optimal pollination extends to approximately two-thirds of horticultural crops in the country [5], particularly enhancing productivity and post-harvest storage qualities in fruits like apples, raspberries, and peaches [3,6].

Australia is renowned for having one of the healthiest honey bee populations globally. This is largely due to its geographic isolation and strict biosecurity measures, which have kept many pests and pathogens away from the mainland [7]. However, beekeepers still need to remain vigilant for established pests and pathogenic diseases [8]. The outbreak of varroa mite in Newcastle in June 2022 highlighted the urgent need for further research to better understand and improve our knowledge of both the pests and pathogens affecting Australian honey bees [9]. While microbial pathogens are indeed present in Australian apiaries, research on their widespread occurrence and transmission remains limited [10]. Understanding how these pathogens, pests, and parasites influence colony productivity and health requires extensive surveillance across various sites in Australia [11]. Identifying the presence of pests or pathogens affecting a hive can be achieved by analysing hive substances, such as wax, pollen, and honey. This is due to the potential transfer of pests, pathogen, and their material during the production of hive substances following exposure while foraging.

These essential pollinators face significant threats from a range of bacterial, viral, and parasitic infections, which compromise their health and productivity. Environmental stressors such as exposure to harmful chemicals, inadequate nutrition, and unsustainable agricultural practices also play a role [12]. American foulbrood (*Paenibacillus larvae*) and European foulbrood (*Melissococcus plutonius*) are brood diseases that compromise larval health and weaken overall colony vitality [13]. Infections caused by fungi, microsporidians, and arthropods contribute to colony collapse, affected by regional genetic variations and additional stressors [14,15].

Molecular methods, owing to their high sensitivity, accuracy, and capability for early detection of pests and pathogens, have been recommended by several researchers over microbial methods for identifying these threats in environmental samples [16]. Polymerase chain reaction-based assays have been developed for the detection of honey bee pests and pathogens from bees and other hive materials [17,18]. End-point PCR is the most widely used method for detecting bacteria (*Paenibacillus larvae*, *Melissococcus plutonius*), microsporidian fungi (*Nosema* spp.), fungi (*Ascosphaera apis*), and arthropods (*Aethina tumida* and *Galleria mellonella*) from honey and other hive samples [19,20,21,22]. Although, arthropods are not true pathogens, their significant pest burden impact honey bee colony health. PCR is highly regarded for its consistent, specific, and precise identification of a multitude of pathogens, making it an essential tool for routine pathogen screening in surveillance operations [23]. Traditionally, adult bees have been the primary source of pathogen detection [24], the evolution of environmental DNA/RNA-based detection techniques presents a superior alternative to direct sampling from hosts [25]. This advancement has prompted numerous research initiatives employing PCR-based assays to detect bee pathogens and parasites directly from honey samples, offering a less invasive yet effective surveillance approach [26].

Monitoring disease in adult bees is challenging due to the resource-intensive requirements and costs associated with collecting data from individual bees. This method provides only a snapshot of disease prevalence at specific times, limiting the ability to track changes over time [27]. In contrast, hive materials such as honey provide a valuable source of environmental DNA (eDNA) for the detection of invasive organisms.

Honey eDNA has, thus, emerged as a valuable resource for monitoring honey bee pathogens, pests, and parasites [19,28]. The genetic material left by organisms in the environment, known as eDNA, is a persistent biomolecular marker that can be collected, extracted, and analysed from various substrates, making it a powerful tool for detecting and monitoring both microbial and macrobial communities effectively [29,30]. Honey’s stable nature, characterized by low water content and acidic pH, preserves remnants of microorganisms, offering insight into historical records of colony pathogens [31]. Recent advances have demonstrated the use of metagenomic and next-generation sequencing (NGS) approaches to characterize microbial communities from honey. These high-throughput methods complement traditional detection methods, offering a promising tool for future surveillance efforts [32,33]. Recognizing honey as a reservoir of exogenous DNA underscores the importance of monitoring and managing bee health, including detecting disease-causing organisms in honey to prevent disease spread among colonies.

In our study, we collected a total of 135 honey samples from diverse botanical and geographical regions spanning different states in Australia. These samples were utilized to isolate eDNA using the bead-beating-silica DNA extraction method and identified pests and pathogens known to impact honey bee populations in Australia. We also examined the co-occurrence pattern of these pests and pathogens in honey samples.

## 2. Materials and Methods

### 2.1. Acquisition of Commercial Honey Samples

A total of 135 honey samples were gathered between 2022 and 2024 from diverse regions across Australia, including trade markets and directly from beekeepers, with regional distribution and sampling location to ensure a comprehensive representation of honey sources (Appendix A) [34]. The sample comprised both poly-floral and mono-floral varieties, which were obtained from different states: Victoria (27), New South Wales (29), Queensland (24), Northern Territory (1), Western Australia (22), South Australia (20, including 6 samples from Kangaroo Island), Tasmania (12). Only one sample was obtained from Northern Territory (2762 registered hives) due to limited shipping options to our lab. Kangaroo Island is famous for its Ligurian honey bee sanctuary. Established in 1885, it preserves the purest strain of *Apis mellifera ligustica* by preventing crossbreeding. Strict biosecurity laws prevent the introduction of other bees and bee products to maintain the genetic purity of the bee population, making it a unique, valuable site for pest and pathogen surveillance.

### 2.2. eDNA Extraction from Honey Samples

DNA was extracted from honey following an in-house extraction protocol, modified from Waiblinger et al. [35] with adjustments, incorporating both pre-treatment and a post-treatment phase as detailed by Soares et al. [36].

For the pre-treatment phase, 50 g of honey was evenly partitioned into four sterile 50 mL conical tubes, each containing 12.5 g of honey. Subsequently, 40 mL of ultrapure water was added to each tube and vortexed until completely homogenized. The samples underwent incubation at 40 °C for 10 min in a water bath with agitation, followed by centrifugation at 4700× *g* for 35 min. The resulting supernatant was discarded, and the pellets were resuspended in 5 mL of ultrapure water and combined with the same honey sample into a single conical tube. This suspension was then subjected to an additional centrifugation at 4700× *g* for 30 min. After centrifugation, the supernatant was removed, and the pellet was resuspended in approximately 500 µL of ultrapure water. Subsequently, the suspension was transferred to a 2 mL tube containing seven glass beads (approximately 5 mm in size) and vortexed for 2 min. The glass beads were removed, and the mixture underwent centrifugation at 11,000× *g* at 4 °C for 10 min. The resulting pellet served as the starting material for DNA extraction.

To each pre-treated sample pellet, 860 µL of a TNE (10 mM Tris-HCL, 150 mM NaCl, 2 mM ethylenediaminetetraacetic acid, 1% (*w*/*v*) sodium dodecyl sulphate, pH 7.5) extraction buffer was added, which was pre-heated at 60 °C. Additionally, 100 µL of 5 M guanidine hydrochloride (*w*/*v*) and 40 µL of proteinase K solution (20 mg/mL^−1^) were added and subsequently vortexed. Following a 3 h incubation at 60 °C in a Thermomixer Comfort (Eppendorf AG, Hamburg, Germany) with agitation at 900 rpm, the suspension underwent centrifugation for 15 min at 17,000× *g* at 4 °C. The supernatant was collected for DNA purification.

### 2.3. eDNA Purification

The resulting extracted DNA was mixed with 2 volumes of 6 M sodium iodide (NaI) and 100 µL of 100 mg/mL silica dioxide (SiO_2_), and binding occurred with gentle agitation on a rocker for 30 min. Subsequently, centrifugation was performed (10 min at 5000× *g* at 4° C), with the supernatant discarded and 500 µL of a silica wash buffer (50% (*v*/*v*) ethanol, 10 mM Tris-HCl, 100 mM NaCl, 1 mM EDTA, pH 8) added to the pellet and resuspended via vortexing. The solution was centrifugated for 1 min at 4700× *g* at 4 °C, with this wash step repeated two more times. Care was taken to remove all the supernatants. To elute the eDNA from the silica matrix, 50 µL of elution buffer (10 mM Tris-HCl, pH 8) was added and incubated at 70 °C for 5 min. Following incubation, the sample was centrifugated for 5 min at 16,000× *g*, with the supernatant collected and stored at −20 °C until further analysis. The concentration of the DNA was measured using a Nanodrop Eight spectrophotometer and visualized by electrophoresis in a 2% (*w*/*v*) agarose gel in a 1× TBE buffer (0.13 M Tris, 45 Mm Boric acid, 2.5 mM EDTA pH 7.6) containing 0.2 µg/mL ethidium bromide (Merck Life Science Pty Ltd., Melbourne, Australia). Successful eDNA extraction and purification was determined by the amplification of *Apis mellifera* mtDNA by end-point PCR.

### 2.4. PCR Analysis

Gene targets and primer sequences were selected based on validated and published studies, as outlined in Table 1. We assessed the presence of both pathogens and pests using singleplex and multiplex PCR, with PCR conditions optimized for each target based on their specific annealing temperature and amplicon size (Table 2). Positive DNA controls for each pathogen and pests were generated by the chemical synthesis of each gene (Table 1) from Integrated DNA Technologies (IDT) and subsequently cloning them into the TOPO vector according to the manufacturer’s instructions. The sensitivity of each primer set was validated through the serial dilution of synthetic plasmid DNA from a 10^−1^ ng/µL to 10^−9^ ng/µL concentration, with the lowest detectable limit (10^−5^ ng/µL) (Appendix A). Specificity was evaluated using genomic DNA of *P*. *larvae*, *M. plutonius*, *A. apis*, *N. apis*, *N. ceranae*, *A. tumida*, and *G. mellonella*, with no cross-reactivity observed (Appendix A). The revalidation of each primer confirmed their high sensitivity and specificity.

PCR reactions were performed in a total volume of 25 µL, containing 1× GoTaq^®^ Green Master Mix (Promega, Madison, WI, USA), 25 ng of template DNA, and species-specific primer concentration. The primer concentrations were optimized as follows: 0.5 µM for *A. mellifera*, *N. apis*, *N. ceranae*, and *G. mellonella*; 0.4 µM for *P. larvae* and *M. plutonius*; 0.3 µM for *A. tumida*; and 0.1 µM for *A. apis*. The thermal cycling conditions were performed as described in Table 2. A negative control, devoid of template DNA, was included in each assay. Subsequently, the amplified DNA fragments were subjected to electrophoresis in a 2% (*w*/*v*) agarose gel in a 1× TBE buffer containing 0.2 µg/mL ethidium bromide. Three confirmed positive samples for each pathogen and pest were sequenced to verify the correct target amplification.

### 2.5. Assessing Changes in Pest and Pathogen Prevalence over Time

To assess changes in the pathogen prevalence over time, we used the data from a study conducted between 2013 and 2015 on 155 apiaries, collecting approximately 25 adult bees per hive and pooling them to analyse pathogen prevalence at the apiary level in Australia as a reference point [8]. While our study utilised honey-derived eDNA, we evaluated potential shifts in the prevalence and distribution of these biological threats across the two time periods, supported by a bar diagram illustrating the changes.

### 2.6. Statistical Analysis

The diversity of pathogens and pests detected in each honey sample was quantified and visualized using the ggplot2 package in R [37]. To assess the patterns of pathogen and pest co-occurrence across the samples, we utilized the R package co-occur (v. cooccur_1.3.tar.gz) [38], which employs a probabilistic model to evaluate species co-occurrence [39]. This model includes combinatorial methods that calculate the probability of the observed frequency of co-occurrence (p_gt). A p_gt value significantly less than expected (p_gt < 0.05) indicates a positive association (+ve), with p_gt > 0.05 indicating negative association (−ve) or independent co-occurrence (p_lt > 0.05), which indicates no evidence of species interaction and suggests no significant association. The effect size, which ranges from −0.02 to 0.08, indicates the strength of co-occurrence (+ve/−ve) between two species, either pathogen or pest present in a sample, and is assessed using the probability matrix (p_gt and p_lt). In total, 21 species pairs were analysed to determine the frequency of their co-occurrence across the samples and the nature of their association (+ve or −ve).

**Table 1 insects-16-00764-t001:** PCR primers used in this study to amplify eDNA extracted from honey samples.

**Target Species**	**Primer Name ^1^**	**Accession No.**	**Primer Sequence (5′-3′)**	**Amplified Region**	**Product Size (bp)**	**Reference**
Singleplex PCR						
*Apis mellifera*	AM Forward AM Reverse	EF033649.1	GGCAGAATAAGTGCATTG TTAATATGAATTAAGTGGGG	mtDNA COI-COII	C 85, M 138 ^2^	[40]
*Nosema apis*	Nose_apis_chen_F Nose_apis_chen_R	U97150.1	CCATTGCCGGATAAGAGAGT CCACCAAAAACTCCCAAGAG	SSUrRNA	269	[41]
*Nosema ceranae*	Nose_cera_chen_F Nose_cera_chen_R	DQ486027.1	CGGATAAAAGAGTCCGTTACC TGAGCAGGGTTCTAGGGAT	SSUrRNA	250	[41]
*Aethina tumida*	Atum-3F Atum-3R	MF943248.1	CCCATTTCCATTATGTWYTATCTATAGG CTATTTAAAGTYAATCCTGTAATTAATGG	COI	97	[42]
*Galleria mellonella*	GallMelCox1-F GallMelCox1-R	KT750964.1	TGAACTTGGTAATCCTGGTTCT TATTATTAAGTCGGGGGAAAGC	COI	182	[42]
Multiplex PCR						
*Paenibacillus larvae* *Melissococcus plutonius* *Ascosphaera apis*	Han233PaeLarv16S_F Han233PaeLarv16S_R Mp_Arai187_F Mp_Arai187_R AscosFORa AscosREVa	NZCP019687.1 AB778538.1 U68313.1	GTGTTTCCTTCGGGAGACG CTCTAGGTCGGCTACGCATC TGGTAGCTTAGGCGGAAAAC TGGAGCGATTAGAGTCGTTAGA TGTGTCTGTGCGGCTAGGTG GCTAGCCAGGGGGGAACTAA	16S rRNA NapA 18S rRNA	233 187 136	[43]

^1^ The internal name of the forward and reverse primers. ^2^ The size of the amplified fragment may vary depending on the mitochondrial lineage, C, which is highly frequent in *A.m. lingustica*, and M in *A.m. mellifera* (C or M).

**Table 2 insects-16-00764-t002:** Optimized PCR conditions for detecting pests and pathogens.

**Target Species**	**Steps**	**Optimized Conditions**	**Time**	**Cycle**
Multiplex PCR				
*P. larvae* *M. plutonius* *A. apis*	Initial Denaturation Annealing Extension Final Extension	95 °C 95 °C 63 °C 72 °C 72 °C	2 min 1 min 1 min 1 min 5 min	35×
Single plex PCR				
*N. apis* *N. ceranae*	Initial Denaturation Annealing Extension Final Extension	95 °C 94 °C 58.6 °C 68 °C 72 °C	2 min 15 s 30 s 1 min 7 min	35×
*A. tumida*	Initial Denaturation Annealing Extension Final Extension	95 °C 98 °C 54 °C 72 °C 72 °C	3 min 20 s 30 s 1 min 7 min	35×
*G. mellonella*	Initial Denaturation Annealing Extension Final Extension	95 °C 98 °C 61 °C 72 °C 72 °C	3 min 1 min 1 min 1 min 1 min	35×

## 3. Results

### 3.1. Assessment of Extracted DNA

Before attempting to analyse for the presence or absence of honey bee pathogens and pests, we optimized PCR for the amplification of various targets using either single or multiplex PCR [41,42,43]. To verify the successful extraction and purification of eDNA from all honey samples, the presence of the *Apis mellifera* mtDNA was determined by PCR [40]. The amplification resulted in fragments of 85 base pairs (bp), specific to the *A. mellifera* C lineage prevalent in *A. m. lingustica* and 138 bp specific to the M lineage, characteristic of *A. m. mellifera* (Figure 1). Successful PCR amplification was achieved for all tested honey samples, indicating the successful extraction and purification of eDNA from the honey.

Every honey sample underwent PCR analysis to detect various common pathogens and pests that affect honey bees (Appendix A). A representative example of data from samples 104 to 110 is shown in Figure 1A, and that for samples 125 to 131 in Figure 1B. Samples 107 and 108 were positive for *N. apis*, while samples 105–110 tested positive for *N. ceranae* (Figure 1A). In the multiplex PCR targeting bacterial pathogens *P. larvae* and *M. plutonius*, and fungal pathogen *A. apis*, sample 107 amplified all three pathogens, whereas sample 108 tested positive for *P. larvae* and *M. plutonius* (Figure 1A). No amplification was detected in samples 104, 105, 106, 109, and 110 (Figure 1A). Furthermore, samples 125–131 indicated the presence of the arthropod pests *A. tumida* and *G. mellonella* (Figure 1B). All samples tested positive for *A. tumida* except samples 126 and 130, which tested positive for *G. mellonella* (Figure 1B).

### 3.2. Prevalence Pattern Across Different Australian States

Analysis of 135 honey samples collected from various locations in Australia revealed *N. ceranae* emerged as the most prevalent pathogen, present in 57% of the samples. This was followed by the pests *A. tumida* (40%) and *G. mellonella* (37%), and the pathogens *P. larvae* (21%), *N. apis* (19%), and *M. plutonius* (18%). *A. apis* was detected in a smaller proportion of the samples, with a prevalence of 5%. Additionally, 25 samples (19%) tested negative for all pathogens and pests analysed. A comparison between the current data and the most recent national survey of honey bee pathogens and pests, conducted in 2014 [8], reveals a significant shift in their levels and distribution across Australia over the subsequent seven years (Figure 2). The detailed geographical distribution of bacterial, fungal pathogens, and arthropod pests throughout Australia over the two years is summarized in Table 3, Table 4 and Table 5.

Regional variations in pest and pathogen prevalence were evident across different Australian states. Specifically, American foulbrood (AFB), caused by *P. larvae* and European foulbrood (EFB), caused by *M. plutonius*, were identified in Victoria (VIC), New South Wales (NSW), Queensland (QLD), and Tasmania (TAS) (Table 3). Notably, VIC exhibited a high prevalence, with 33% of the sample testing positive for *P. larvae* and 37% for *M. plutonius*. In NSW and QLD, there was no significant difference in the prevalence of *P. larvae* (28% and 21%, respectively) and *M. plutonius* (14% and 13%, respectively). TAS had a prevalence of 25% for both *P. larvae* and *M. plutonius*. In contrast, Western Australia (WA) only harbors *P. larvae* (14%) and is free from *M. plutonius*, while South Australia (SA) exclusively hosts *M. plutonius* (29%), with *P. larvae* absent. Similarly, *M. plutonius* was absent in WA and Kangaroo Island (KI) (Table 3).

The primary fungal disease impacting honey bees, *A. apis* (commonly referred to as chalkbrood), was detected at low levels in Victoria (4%) and New South Wales (7%), with no significant presence in Queensland, Northern Territory, Western Australia, and South Australia (Table 4). In contrast, Tasmania exhibited a relatively high incidence of *A. apis* at 33% (Table 4).

*Nosema ceranae* was frequently found across all states, with a particularly high prevalence in VIC (70%), followed by SA (64%), TAS (50%), and WA (45%). Remarkably, even a single sample from NT tested positive for *N. ceranae* (Table 4). The prevalence rates in NSW (59%) and QLD (58%) were similar, suggesting no apparent difference based on the identification of positive cases and calculated percentage. *N. apis* was identified in TAS (58%), WA (23%), VIC (19%), SA (14%), and NSW (10%), but it was not detected in QLD or NT (Table 4). Additionally, 16% of the samples tested positive for both *N. ceranae* and *N. apis*, indicating co-occurrence (Table 4).

The invertebrate pests of honey bees, *A. tumida* and *G. mellonella*, were present in honey samples from all states except KI (Table 5). *A. tumida* showed a high prevalence in QLD (71%), SA (57%), VIC (56%), TAS (33%), and NSW (31%), while WA had a significantly lower prevalence at 5%. On the other hand, *G. mellonella* was most prevalent in TAS (84%) followed by SA (57%), WA (45%), VIC (44%), QLD (25%), and NSW (10%). Notably, in SA, 57% of the samples tested positive for both *A. tumida* and *G. mellonella* (Table 5).

### 3.3. Pest and Pathogen Prevalence on Kangaroo Island

Six honey samples collected from the commercial beekeepers on KI were tested for bacterial, fungal pathogens, and arthropod pests. The results were significant, as all samples tested negative for the three brood disease agents (*P. larvae*, *M. plutonius*, and *A. apis*), as well as the pests *A. tumida* and *G. mellonella*. The only honey bee pathogens detected on KI were the fungal pathogens *N. apis* and *N. ceranae*, with 50% and 17% of samples testing positive, respectively (Table 4).

### 3.4. Trends in Co-Occurrence of Pathogens and Pests

This study uncovered a diverse array of honey bee pathogens and pests present in individual honey samples, highlighting a significant degree of co-occurrence across the surveyed samples. The analysis revealed that 30% (40/135) of the samples contained a single type of pest or pathogen, while 20% (27/135) displayed two distinct types. Co-occurrence involving three and four different pests or pathogens was identified in 13% (18/135) and 10% (13/135) of the samples, respectively. Furthermore, 6% (8/135) contained five different types, with fewer than 4% (6/135) showing six distinct pests or pathogens. Notably, none of the samples tested positive for all seven types analysed (Figure 3A).

The probabilistic co-occurrence model revealed several significant positive associations (Figure 3B). Among these, positive co-occurrences were observed between *P. larvae* and *M. plutonius* (p_gt = 0.00001, p_lt = 1.00, effect size = 0.0666). Notably, *M. plutonius* exhibited strong to moderate positive co-occurrence with all studied pests and pathogens, including a moderate association between *A. apis* (p_gt = 0.00198, p_lt = 0.99, effect size = 0.0281), *N. apis* (p_gt = 0.02280, p_lt = 0.99, effect size = 0.0311), and *N. ceranae* (p_gt = 0.02776, p_lt = 0.99, effect size = 0.0348), and a strong positive association with *A. tumida* (p_gt = 0.00082, p_lt = 0.99, effect size = 0.0548) and *G. mellonella* (p_gt = 0.00017, p_lt = 0.99, effect size = 0.0614). Additionally, *A. apis* displayed weak positive co-occurrence with *N. apis* (p_gt = 0.02949, p_lt = 0.99, effect size = 0.0192) and *G. mellonella* (p_gt = 0.00931, p_lt = 0.99, effect size = 0.0259), while *N. apis* showed a moderate positive association with *N. ceranae* (p_gt = 0.00752, p_lt = 0.99, effect size = 0.0444) and *G. mellonella* (p_gt = 0.01889, p_lt = 0.99, effect size = 0.0385). *N. ceranae* had a strong positive association with *A. tumida* (p_gt = 0.00120, p_lt = 0.99, effect size = 0.0666) and a moderate positive association with *G. mellonella* (p_gt = 0.00407, p_lt = 0.99, effect size = 0.0577). A strong positive co-occurrence was also observed between the two arthropod pests (p_gt = 0.00058, p_lt = 0.99, effect size = 0.0696).

Among the 21 pairs examined, including pest–pest, pathogen–pathogen, and mixed pest–pathogen combinations, 13 pairs exhibited positive associations, ranging from weak to strong, and tended to co-occur more frequently than expected (Appendix A). In contrast, eight pairs did not display statistically significant positive or negative associations, suggesting no evidence of interaction between the involved pests and/or pathogens. For instance, the interaction between *P. larvae* and *N. ceranae* is characterized by a weak negative association (p_gt = 0.67507, p_lt = 0.48, effect size = −0.0044), indicating that the two pathogens are likely independent of each other. Similarly, *N. apis* and *A. tumida* exhibited a weak negative association (p_gt = 0.28640, p_lt = 0.84, effect size = −0.0133). Additionally, *P. larvae* showed no positive or negative association with *A. apis* (p_gt = 0.15581, p_lt = 0.96, effect size = 0.0111), *N. apis* (p_gt = 0.06581, p_lt = 0.97, effect size = 0.0251), *A. tumida* (p_gt = 0.28489, p_lt = 0.84, effect size = 0.0133), or *G. mellonella* (p_gt = 0.15120, p_lt = 0.92, effect size = 0.0207). Similarly, *A. apis* did not exhibit significant co-occurrence with *N. ceranae* (p_gt = 0.62275, p_lt = 0.67, effect size = 0.0007) or with *A. tumida* (p_gt = 0.28532, p_lt = 0.90, effect size = 0.0088), showing that these organisms occur independently of each other, with no strong evidence of biological interaction.

## 4. Discussion

Honey serves as an ideal source of eDNA because foraging honey bees collect and transfer environmental microorganisms and contaminants to the hive, depositing them in the honeycomb and honey [26,44]. This eDNA reveals the presence of various organisms, including bee pathogens and pests, making honey a valuable non-invasive tool for monitoring colony health and detecting invasive species [19,33].

In this study, we aimed to investigate the exposure of managed honey bee populations to potential pests and pathogens across different geographic regions within Australia. We used honey-based sampling for apiary-level surveillance to understand regional patterns of pest and pathogen presence. To achieve this, we expanded existing end-point PCR assays to detect seven key disease-causing organisms relevant to honey bee health: *Paenibacillus larvae*, *Melissococcus plutonius*, *Ascosphaera apis*, *Nosema apis*, *Nosema ceranae*, *Aethina tumida*, and *Galleria mellonella*. While several studies have developed end-point PCR-based diagnostic assays to identify these pests and pathogens [19,42,43,45,46], this approach has not yet been applied within Australia in recent years. The only prior surveillance effort, conducted between 2013 and 2015, relied on adult bees rather than honey samples [8]. Results from that period showed a higher prevalence of *N. apis*, *N. ceranae*, and *A. apis*, as well as moderate levels of *A. tumida* (small hive beetle—SHB) and *G. mellonella* (wax moth), with a lower prevalence of *P. larvae* and *M. plutonius*.

In comparison, our current data highlight a shift in prevalence, with *N. ceranae*, *A. tumida*, and *G. mellonella* emerging as the most widely distributed and frequently detected pathogens and pests across Australia. Furthermore, the bacterium *P. larvae*, and *M. plutonius*, continue to be detected at low prevalence, while the microsporidian *N. apis* and fungus *A. apis* showed reduced prevalence compared to the previous data. These findings reveal novel trends in the distribution of pests and pathogens, supporting the notion to continue monitoring key bee disease hotspots in Australia.

In this study, *N. ceranae* showed the highest prevalence among the *Nosema* species and is known as a globally distributed honey bee disease [47]. *N. ceranae* was initially believed to be restricted to Asian honey bees (*Apis ceranae*) when first detected in Beijing, China, in the 1990s [48]. At that time, *A. mellifera* was believed to be susceptible only to *N. apis*. However, this view changed when *N. ceranae* was identified in *A. mellifera* using naturally infected honey bee isolates from various regions worldwide [49,50]. This host shift of *N. ceranae* from *A. ceranae* to *A. mellifera* has occurred on a global scale, raising questions about the factors driving this transition, but it is likely due to the movement of infected honey bees through the increasing global commercial trade [51,52,53,54].

Our findings show that *N. ceranae* has largely dominated over *N. apis* in Australia, reflecting a broader global trend [16,55,56,57,58,59,60,61]. However, the overall prevalence of *N. ceranae* and *N. apis* detected in our honey samples was lower than the levels reported in adult bee samples from a previous Australian study conducted between 2013 and 2015 [8]. This difference is likely because of the type of samples analysed. Active *Nosema* infections occur in the gut epithelial cells of bees, making them more readily detected in bee tissue than in hive-derived materials like honey [62,63,64]. Therefore, the lower prevalence observed in honey samples may not necessarily reflect a reduced infection rate but rather the limitation of honey as a diagnostic sample for active *Nosema* infection.

Many studies have suggested that a shift in pathogen prevalence may be influenced by climatic factors, as *N. ceranae* thrives in warmer environments, which may explain its widespread presence across much of Australia [65,66]. In contrast, *N. apis* remains viable even under freezing conditions and is, therefore, more prevalent in colder regions [67]. These temperature preferences may explain *N. ceranae*’s increasing dominance, and with global warming continuing to raise the baseline temperature, *N. ceranae* is likely to expand further within the Australian bee population [68]. Furthermore, *N. ceranae* has higher virulence, characterized by increased honey bee mortality compared to *N. apis* [66]. Combined with its spore tolerance to temperatures up to 60 °C and resistance to desiccation, this supports its capacity for rapid prevalence expansion. If left untreated, infected colonies often collapse, and *N. ceranae* has been linked to unexplained colony losses worldwide, emphasizing the need for effective control strategies such as queen replacement, the use of disinfectants (ammonia solution or sodium hypochlorite solution) to reduce spore viability, and organic acids treatments (organic acids or essential oils) as demonstrated by Formato et al. [69,70].

In KI, the honey bee species *A. mellifera ligustica* has been preserved in genetic isolation through strict biosecurity measures aimed at preventing hybridization with other bees [71]. The 2013–2015 study also tested pest and pathogen presence on this unique bee population in KI and reported the presence of *A. apis*, *G. mellonella*, *N. apis*, and *N. ceranae* in adult bees [8]. However, our analysis of honey samples detected only *N. apis* and *N. ceranae*, suggesting the absence of other pests and pathogens. This likely reflects a combination of strong biosecurity enforcement and the limited pests and pathogen diversity observed in the small number of samples collected from KI. While only six honey samples were analysed from this region—introducing potential sampling bias, these findings, along with continued surveillance and legislative support, underscore the role of ongoing monitoring in preserving the health and resilience of the unique honey bee population on KI.

The chalkbrood agent, *A. apis*, was detected in 5% of the analysed honey samples, primarily from VIC and NSW. Globally, chalkbrood is distributed across major beekeeping regions, including Central America, North America, Mexico, Chile, Japan, China, Turkey, Africa, and the Philippines [72]. In Australia, *A. apis* was first reported in QLD and subsequently spread across all states [73,74]. While our study found only a low prevalence of *A. apis* in honey samples, previous work by Robert et al. [8] identified chalkbrood mummies in 66% of hives across Australia, primarily along the eastern coast, with a single case in WA between 2013 and 2015. Notably, *A. apis* was not detected in honey samples from WA in our study. The higher prevalence reported in 2013–2015 in Australia could be due to the collection of brood visibly affected by disease, whereas our study offers an unbiased snapshot based on random honey sampling. Differences in the results could also be due to fluctuations in outbreaks, dormant spores, or spore concentrations below the detection level. Guimarães-Cestaro et al. [75] demonstrated that *A. apis* can be detected in honey samples at concentrations as low as 7.5 spores/mL using a multiplex PCR detection method targeting the internal transcribed spacer regions (ITS1 and ITS2), which differs from our approach that targeted the 18S rRNA gene. While our sample source of the hive material honey could be limiting the detection of some pests and pathogens, our data does highlight that honey can still capture residual spores and eDNA, supporting its use as a non-invasive, non-bias method for surveillance at the hive level.

The prevalence of *P. larvae* and *M. plutonius*, the causative agents of AFB and EFB, detected in this study (21% and 18%, respectively) was notably higher than the 9% (*P. larvae*) and 5% (*M. plutonius*) reported by Robert et al. [8] who sampled symptomatic hives. The variation in the prevalence of both bacterial pathogens over time could be due to the broader sampling approach, which detects early-stage infections rather than sampling diseased broods. The persistence of *P. larvae* spores in honey, which remain viable for 15 years and are capable of germinating under favourable conditions, could also contribute to its continued presence, even in the absence of visible symptoms [75]. However, the rise in the prevalence of *M. plutonius* (a non-spore-forming bacteria) potentially indicates an increase in colony stress linked to factors such as pesticide exposure, climate change, or poor nutrition [76]. Geographically, our findings align with previous reports indicating that WA, NT, and KI remain free from *M. plutonius*, and this consistency suggests effective regional biosecurity measures [8,77].

These findings reflect a broader global trend, with diverse patterns of *P. larvae* and *M. plutonius* prevalence reported across different regions. In Indonesia, *P. larvae* was not detected, but *M. plutonius* was identified for the first time from honey and worker bee samples using end-point PCR [16]. Ribani et al. [19] reported a high prevalence rate of *M. plutonius* (87%) and *P. larvae* (49%) in Italian honey samples processed between 2004 and 2018 using species-specific primers and end-point PCR. Similarly, *P. larvae*, DNA and spore were detected in 40% of the analysed honey samples in Italy by end-point PCR and spore count, indicating a long-term persistence of the disease in hive products [78]. A study in the U.S. reported that *M. plutonius* (19.2%) was more common than *P. larvae* (8.6%) in symptomatic colonies despite the resilience and ease of spread of *P. larvae* spores. These results were based on visual inspection, the microscopic examination of swabs from infected hives, and the culturing of *P. larvae* spores in brain heart infusion with thiamine [76].

Globally, *P. larvae* are present in all beekeeping countries and are classified by the World Organization for Animal Health (WOAH) as a highly dangerous infectious disease in animals. First identified in North America using culture-based methods and brood inspections, *P. larvae* are known to produce large quantities of long-lived spores, which are essential for infection [79]. Once a colony is infected, recovery is not possible, often requiring the destruction of the hives and equipment to prevent spread [80,81]. To support the early detection and prevent hive incineration, Ackerly et al. [82] developed a highly sensitive and field-deployable LAMP assay for the rapid identification of *P. larvae*, facilitating timely and effective disease management. Whereas *M. plutonius* is less contagious than *P. larvae*, it has expanded beyond Europe and North America to Africa, South America, India, Japan, and Australia [82]. While many countries rely on antibiotic treatment for *P. larvae* (tetracycline or tylosin) and *M. plutonius* (oxytetracycline), the overuse of these antibiotics has led to global concerns about antibiotic resistance [76]. In contrast, Australia maintains a restricted-use approach, with a complete prohibition on antibiotic treatment for *P. larvae* and permitted use only under prescription for *M. plutonius*, combined with proactive management and strong biosecurity measures, which could contribute to the lower disease burden observed nationally [83].

This study highlights the widespread presence of two major arthropod pests, *A. tumida* (the small hive beetle—SHB) and *G. mellonella* (wax moth), in Australian honey samples, with prevalence rates of 40% and 37%, respectively. These findings indicate a significant pest burden, aligning with global patterns [84]. They are widely distributed in all continents, except Antarctica [85,86]. A comparison with earlier Australian data (2013–2015), based on adult bee samples, reveals a current upward trend in both arthropod pests. SHB, once confined to QLD and not found in SA, WA, TAS, and KI, is now showing signs of spreading to previously unaffected regions. In contrast, the wax moth was already present in all states [2].

In South Africa, 69% of apiaries tested positive for SHB and wax moth infestation between 2010 and 2011 [79], while in Nigeria, recent surveys report an infestation rate of 21% for SHB and 5% for wax moths [87]. This suggests a consistent global challenge in managing these pests, particularly in regions where colonies are already stressed by other factors. Arthropod pests not only cause direct damage but also act as vectors for viral and microbial pathogens, compounding their impact on colony health [88,89,90]. This potential spread of both pests could have profound implications for pest management and biosecurity control, as it suggests evolving pest dynamics may outpace existing strategies. Strengthening colony health, fumigating equipment, and maintaining good hive hygiene are essential Integrated Pest Management (IPM) practices but may need to be adapted in response to shifting prevalence patterns [91].

While this study focused on the detection of SHB and wax moth, other significant parasitic mites that pose serious threats to honey bee health globally—*Varroa destructor*, *Tropilaelaps clareae*, and *Acarapis woodi* were not targeted. Their exclusion represents a limitation, especially given their global spread and potential introduction into Australian apiaries through trade and hive movement [92,93]. Future research should expand surveillance to include these mites using eDNA tools [19]. These findings reinforce the importance of adopting IPM strategies in modern apiculture [87,91]. Expanding IPM to include regular eDNA monitoring could further enhance early detection and targeted response, especially as pests are expanding their distribution.

Since hive health is influenced by multiple factors, understanding co-occurrence between different pests, pathogens, and parasites is critical, as such interactions often contribute to colony collapse [94,95]. However, investigations into co-occurrence in bee hives are limited. Even when co-occurrence is detected in honey samples, it does not necessarily indicate that the same pests and pathogens coexist within individual colonies, as honey may be sourced from multiple hives. Our data, which determine pathogen co-occurrence at the apiary level, reveal a strong to moderate positive co-occurrence of *M. plutonius* with all the tested pests and pathogens, consistent with findings by Deutsch et al. [57], which linked *M. plutonius* to *A. apis* and *N. ceranae*. Aglagane et al. [96] similarly, reported higher prevalence levels of *N. ceranae* and *M. plutonius* in migratory beehives compared to stationary colonies. A strong positive association between *P. larvae* and *M. plutonius* was reported by Sturtevant et al. [97] based on samples collected from the same comb within diseased colonies, indicating the co-occurrence of both bacterial species within individual hives, potentially coinfecting the same brood frames. This strong positive co-occurrence may be explained by the impact of *P. larvae* on overall health, as it compromises immune defence and creates an environment for *M. plutonius* to be established more readily [98].

In addition, our study supports previous findings from 32 Kenyan apiaries showing a significant positive correlation between *A. tumida* and *N. ceranae* [99]. One possible explanation is that *A. tumida* contributes to the biological transmission of *N. ceranae* spores through contact with infected bee faeces, suggesting its potential role as a passive carrier of pathogens [90,99]. Alternatively, *N. ceranae*-infected colonies are often immunocompromised, reducing their ability to defend against secondary threats such as *A. tumida* [100]. Our co-occurrence data further showed a strong positive association between SHB and wax moths, which are frequently found together at the apiary level. The presence of multiple arthropod species within the same hive environment may exacerbate colony stress and disease susceptibility. These patterns suggest complex biological interactions, potentially driven by resource competition, or shared transmission routes [57,101]. The observed trends among pest–pathogens, pest–pests, and pathogen–pathogens indicate possible synergistic or competitive interactions that may intensify disease impact and challenges in colony health. However, these associations do not confirm that these pathogens directly cause infections or facilitate the spread of each other.

As the honey samples obtained in this study were pooled from multiple hives, the results represent an apiary-level snapshot rather than colony-specific data, limiting the detection of active infection or co-occurrence patterns at the individual hive level. Furthermore, the exclusion of certain key arthropod pests and parasites, along with the inclusion of only one sample from the NT, further limits the comprehensiveness of our surveillance and regional inference. While honey preserves genetic material due to its antimicrobial properties, high sugar content, and low water activity, which together inhibit enzymatic degradation, protecting DNA over time, the presence of DNA alone in honey samples does not confirm active infections or infestations but still represents a potential transmission risk [102]. These limitations highlight the need for broader, large-scale surveys that incorporate diverse pathogen and pest targets to understand emerging threats and mitigate the potential disease spillover to other pollinator species.

## 5. Conclusions

For the first time, an initial pest and pathogen prevalence survey using honey samples collected from different states of Australia revealed that *N. ceranae* is the most prevalent pathogen, followed by the pests *A. tumida* and *G. mellonella*, and the pathogens *P. larvae*, *N. apis*, *M. plutonius*, and *A. apis*. The detection of nosemosis on Kangaroo Island highlights emerging biosecurity threats. While honey provides a non-invasive method for monitoring, it limits the interpretation of active infections or hive-level dynamics. More research is required to understand the co-occurrence pattern observed in this study. Future research should expand on conducting routine surveillance every 3–5 years and further developing these assays into a rapid, field-deployable tool to support early detection at the hive level to enhance biosecurity responses.

## Figures and Tables

**Figure 1 insects-16-00764-f001:**
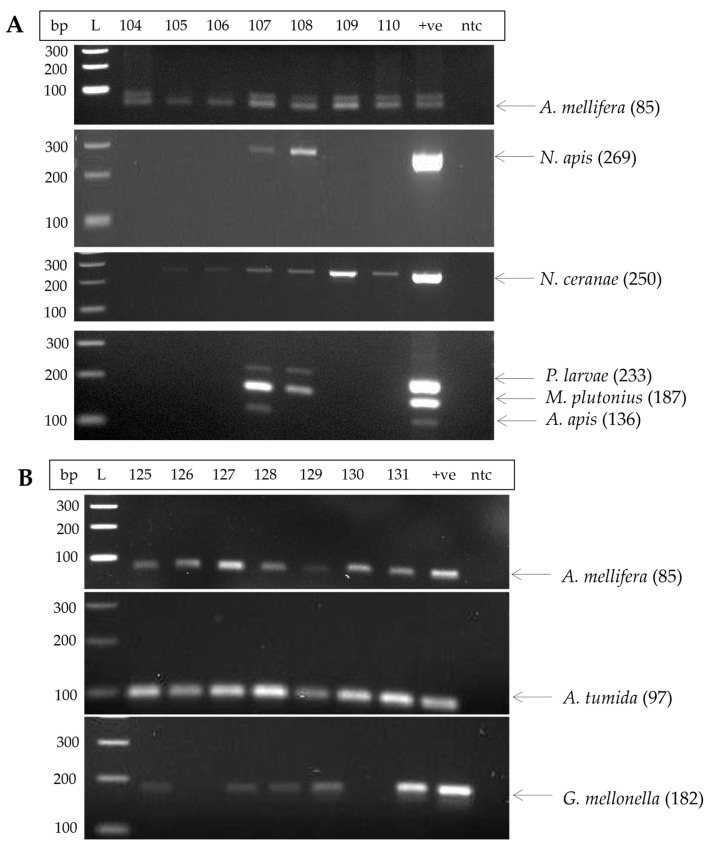
Representative agarose gel electrophoresis of PCR products amplified from honey eDNA by specific primer pairs for honey bee pathogens and pests. (**A**) PCR products amplified for the *A. mellifera*, *P. larvae*, *M. plutonius*, *A. apis*, *N. apis*, and *N. ceranae* from honey samples 104 to 110, with the corresponding size shown in brackets. (**B**) PCR products amplified for the *A. mellifera*, *A. tumida*, and *G. mellonella* from honey samples 125 to 131, with the corresponding size shown in brackets. The lanes included a 100-base pair (bp) DNA ladder (L), a synthetic plasmid containing the gene target of the test pathogen or pest as a positive control (+ve), and nuclease-free water as no template control (ntc).

**Figure 2 insects-16-00764-f002:**
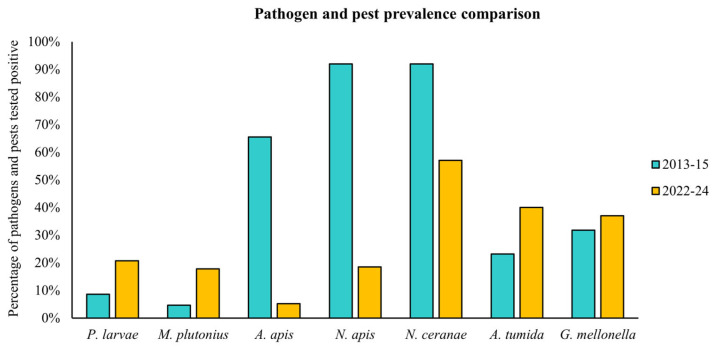
Prevalence of honey bee pathogens and pests surveyed in 2013–2015 [8] and this study (2022–2024).

**Figure 3 insects-16-00764-f003:**
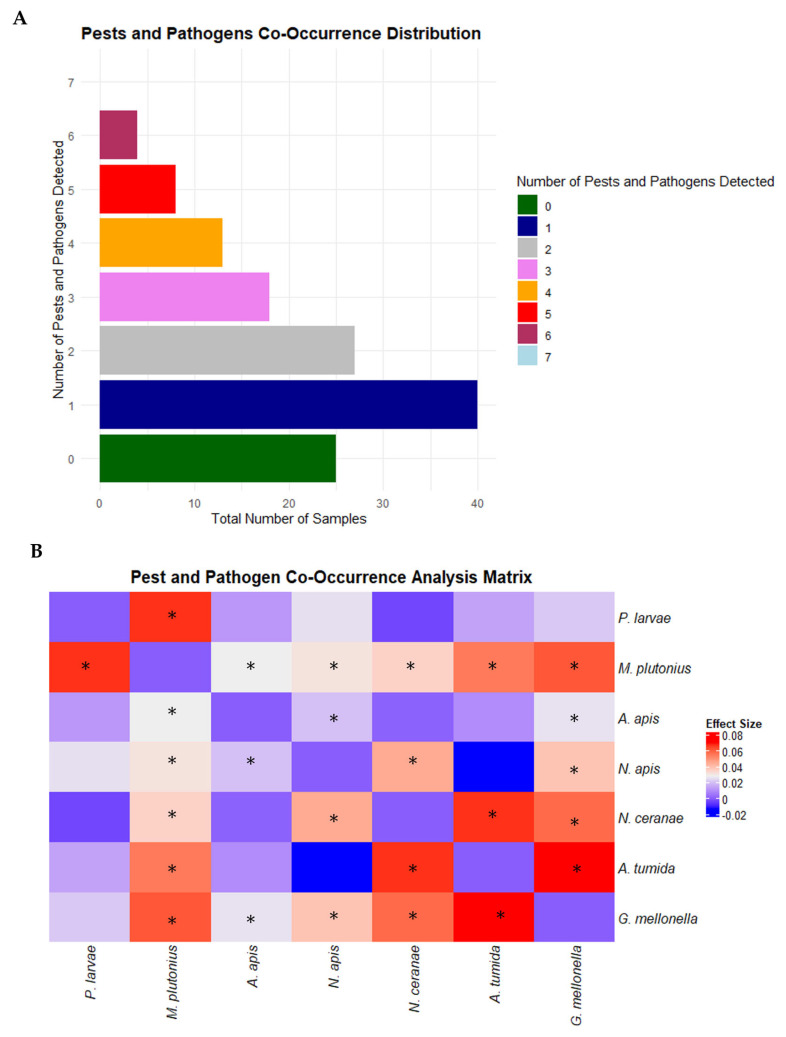
Distribution and co-occurrence of pests and pathogens in a honey sample. (**A**) The bar graph shows the distribution of tested organisms across various honey samples. Each bar represents the number of pests or pathogens detected in the analysed honey samples, enabling the comparison of their frequency among the samples. (**B**) Heat map illustrating co-occurrence patterns among organism pairs. The colour gradient reflects the strength of co-occurrence association among the pests or pathogens (effect size range: −0.02 to 0.08); a darker red signifies a stronger positive co-occurrence, while a darker blue indicates a stronger negative co-occurrence association. This association is evaluated using effect-size metrics (Appendix A). Asterisks (*) indicate that the organisms tend to co-occur more frequently than expected, showing a statistically significant positive association (p_gt < 0.05), with the strength of the association varying from weak to strong depending on the effect size. The heat map is represented by a mirrored imaging pattern (13 pairs).

**Table 3 insects-16-00764-t003:** Distribution of bacterial pathogens *P. larvae* (causative agent of AFB) and *M. plutonius* (causative agent of EFB) across different states in Australia.

State	No. of Samples	No. of Samples Positive for *P. larvae*	% of Positive Samples	No. of Samples Positive for *M. plutonius*	% of Positive Samples
Victoria (VIC)	27	9	33%	10	37%
New South Wales (NSW)	29	8	28%	4	14%
Queensland (QLD)	24	5	21%	3	13%
Northern Territory (NT)	1	0	0%	0	0%
Western Australia (WA)	22	3	14%	0	0%
South Australia (SA)	14	0	0%	4	29%
Kangaroo Island (KI)	6	0	0%	0	0%
Tasmania (TAS)	12	3	25%	3	25%
Total	135	28	21%	24	18%

Abbreviations: VIC—Victoria, NSW—New South Wales, QLD—Queensland, NT—Northern Territory, WA—Western Australia, SA—South Australia, KI—Kangaroo Island, TAS—Tasmania.

**Table 4 insects-16-00764-t004:** Distribution of fungal pathogens of honey bees across different states in Australia.

State	No. of Samples	No. of Samples Positive for *A. apis*	% of Positive Samples	No. of Samples Positive for *N. apis*	% of Positive Samples	No. of Samples Positive for *N. ceranae*	% of Positive Samples	% of Co-Occurrence of *N. apis* and *N. ceranae*
VIC	27	1	4%	5	19%	19	70%	19%
NSW	29	2	7%	3	10%	17	59%	10%
QLD	24	0	0%	0	0%	14	58%	0%
NT	1	0	0%	0	0%	1	100%	0%
WA	22	0	0%	5	23%	10	45%	23%
SA	14	0	0%	2	14%	9	64%	14%
KI	6	0	0%	3	50%	1	17%	17%
TAS	12	4	33%	7	58%	6	50%	42%
Total	135	7	5%	25	19%	77	57%	16%

State abbreviations as defined in Table 3.

**Table 5 insects-16-00764-t005:** Distribution of pests *A. tumida* and *G. mellonella* of honey bee hives across different states in Australia.

State	No. of Samples	No. of Samples Positive for *A. tumida*	% of Positive Samples	No. of Samples Positive for *G. mellonella*	% of Positive Samples
VIC	27	15	56%	12	44%
NSW	29	9	31%	3	10%
QLD	24	17	71%	6	25%
NT	1	0	0%	0	0%
WA	22	1	5%	10	45%
SA	14	8	57%	8	57%
KI	6	0	0%	0	0%
TAS	12	4	33%	10	83%
Total	135	54	40%	49	37%

State abbreviations as defined in Table 3.

## Data Availability

The original contributions presented in this study are included in the article/Appendix A. Raw images for gels can be found at http:/doi.org/ 10.26181/29487338. Further inquiries can be directed to the corresponding author.

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
