# Peer review of "Nationwide Screening for Arthropod, Fungal, and Bacterial Pests and Pathogens of Honey Bees: Utilizing Environmental DNA from Honey Samples in Australia"

_insects, 2025, doi:10.3390/insects16080764_

Round 1

Reviewer 1 Report

Comments and Suggestions for Authors

The authors reported an interesting study that used honey as a source of DNA to monitor the diffusion of pathogens and pests.

Several improvements and corrections are needed before the manuscript can be considered for publication.

1. Title and all over the text, including summary and abstract: the investigated arthropods are not considered pathogens. The correct term is "pests" - all pathogens, parasites and pests can be considered threats or biological threats or something similar. Therefore, more precise definitions are needed. A few examples of words and sentences that should be corrected: lines 13, 14, 15, 27, 28, 29, 34, 35, 47, 52, 55, 75, 77, 163, 164,166, 183, 184, 188, 190, 192, 203, 215, 238, 240, title of figure 2,  287, 291, 300, 301, 302, 304, 308,, 322, 324, 336, 366 titles in figure 4, 338, 339,340,341,342,345,353, 474, 507, and many others.

2. Aethina tumida, and Galleria mellonella are not arthropod parasites, they are pests - a parasite is varroa.

3. Nosema apis and Nosema ceranae are microsporidians, unicellular fungal endoparasites and so on. All definitions of the investigated biological threats should be more precise. 

4. Introduction: line 78: Conventional PCR is end-point PCR.

5. Introduction: lines 86-88. This sentence is in a loop: DNA in honey is first extensively mentioned here , and then it is repeated from lines 94 to 107.  Reading the introduction, it seems the same point is repeated twice and disrupt the flow of the presentation of the main objective of the study. Introduction should be adjusted to avoid this loop. 

6. M&M: line 119; what is the meaning of the asterisk? - in one region, just one honey sample has been collected. The relevance of the information derived is very low in the context.

7. M&M: line 124: The reference Rathinasamy et al. is not appropriate - it does not refer to honey

8. M&M: lines 125-137. It seems that the pre-treatment description derives from another reference not cited here.

9) M&M: line 188: it is not clear what p_gt is

10) Table 1 and line 174: it is not explained why A. mellifera was tested - this is only reported in Results.

11) Table 1: it is not clear what is "suppl" in the correspondence of A. mellifera. There is no accession no. for A. mellifera

12) Lines 209-214: a reference should be cited as this test has been developed by others.

13) lines 239-241: A reference is missed. 

14) Figure 2: How is it possible to compare the results of a study in 2013-2015 with what reported here? This should be elaborated into the text and mentioned in M&M.

15) Tables 3, 4 and 5: Northern Territory: only 1 sample - disclaimers should be clearly reported in the results related to this and discussed.

16) Lines 271-272: what is variance in prevalence? a statistica test should be reported

17) Lines 247-256: acronyms for the various regions should be defined at first use and also reported in Tables 3, 4 and 5 to simplify the reading.

18) Lines 355: end-point PCR (?)

19) Lines 396-388: a reference is missed

20) Lines 414: Chalkbrood agent was detected not Chalkbrood

21) Lines 418-419: Not clear what is referred to, Is it another study? In that case, a reference should be included.

22) Line 429: these acronyms should be defined

23) Line 432-433: italics for P. larvae

24) Lines 442-443: this comment is not clear as it is not reported clearly the comparative analysis between studies.

25) Line 446: Again, the mentioned study reported the presence of the DNA of the bacteria  causing the mentioned diseases. This should be corrected in other parts of the discussion

26) Lines 447-448: not clear the context and this results was obtained in the mentioned study. This should be clarified.

27) Lines 465-466: significant positive trend?

28) Discussion: there is no mention to the fact that the co-occurence is detected on honey samples; this does not mean that the co-occurence is also at the colony level as honey can derived from more colonies and apiaries. This should be clearly mentioned.

29) Some limitations of the study should be reported: the low number of samples tested, the need to include additional pathogens or parasites (for example varroa) and the need to georeference the samples to have a more precise map of the infected regions.

30) References. Another important study that analysed honeyeDNA to monitor the diffusion of a parasite is not mentioned and discussed. This study is relevant for the epidemiological information derived from honey eDNA: https://doi.org/10.1016/j.jip.2021.107628

Author Response

The authors reported an interesting study that used honey as a source of DNA to monitor the diffusion of pathogens and pests.

Several improvements and corrections are needed before the manuscript can be considered for publication.

  1. Title and all over the text, including summary and abstract: the investigated arthropods are not considered pathogens. The correct term is "pests" - all pathogens, parasites and pests can be considered threats or biological threats or something similar. Therefore, more precise definitions are needed. A few examples of words and sentences that should be corrected: lines 13, 14, 15, 27, 28, 29, 34, 35, 47, 52, 55, 75, 77, 163, 164,166, 183, 184, 188, 190, 192, 203, 215, 238, 240, title of figure 2,  287, 291, 300, 301, 302, 304, 308,, 322, 324, 336, 366 titles in figure 4, 338, 339,340,341,342,345,353, 474, 507, and many others.

Response: Thank you for pointing this out, and we agree with this comment. Changes have been made to the table, images whole manuscript  in the following lines

Line 3 Title replaced with Honey Bee pathogens and Pests

Line 13 replaced with bacteria, fungi, pathogens and arthropod pests

Line 16 replaced with bacteria, fungi, and arthropod pathogens and pests affecting bees.

Line 27-28 replaced with “In this study, we extracted eDNA from 135 honey samples and tested for the presence of DNA for seven key honey bee pathogens, pests and parasites”

Line 29- 31 Melissococcus plutonius (bacterial pathogens), Nosema apis, Nosema ceranae (microsporidian fungi), Ascosphaera apis (fungal pathogen), Aethina tumida, and Galleria mellonella (arthropod pests)

Line 35 replaced with for all pathogens and pests.

Line 37 replaced with fungal, and bacterial pathogens and pests affecting honey bees in Australia.

Line 48 -49 replaced with research to better understand and improve our knowledge of both the pests and pathogens affecting Australian honey bees

Line 51 replaced with Understanding how these pathogens, pests, and parasites influence colony .

Line 53 -54 (the whole statement is edited for grammatical correction) has been replaced  with “Identifying the presence of pests or pathogens affecting a hive can be achieved by analysing hive substances such as wax, pollen, and honey.”

Line 55-56 replaced with transfer of pests, pathogen, and their material during the production of hive substances following exposure while foraging.

Line 75-77 replaced with Molecular methods, owing to their high sensitivity, accuracy, and capability for early detection of pests and pathogens, have been recommended by several researchers over microbial methods for identifying these threats in environmental samples.

Line 77- 79 replaced with Polymerase chain reaction-based assays have been developed for the detection of honey bee pests and pathogens from bees and other hive materials.

Line 96 replaced with Honey eDNA has thus emerged as a practical method for monitoring honey bee pathogens, pests and parasites .

Line 104-105 is replaced with monitoring and managing bee health, including detecting disease-causing organisms in honey to prevent disease spread among colonies. 

Line 109 is replaced with “pests, pathogens and parasites known to impact honey bee populations in Australia. We ”

Line 113 replaced with “examined the co-occurrence pattern of these pests and pathogens in honey samples.”

Line 171-173 replaced with “We assessed the presence of both pathogens and pests using singleplex and multiplex PCR, with PCR conditions optimized for each target based on their specific annealing temperature and amplicon size”.

Line 173-176 replaced with “Positive DNA controls for each pathogen and pests were generated by chemical synthesis of each gene (Table 1) from Integrated DNA Technologies (IDT) and subsequently cloning them into the TOPO vector according to the manufacturer's instructions.”

Line 188 replaced with positive samples for each pathogen and pest were sequenced to verify the correct target

Line 191-192 replaced with “The diversity of pathogens and pests detected in each honey sample was quantified and visualized using the ggplot2 package in R ”

Line 192-193 “To assess the patterns of pathogen and pest co-occurrence across the samples, we utilized the R package cooccur ”

Line 198 replaced with “which indicates no evidence of species interaction”

Line 199-203 replaced with The effect size, which ranges from -0.02 to 0.08, indicates the strength of co-occurrence (+ve/-ve) between two species, either pathogen or pest present in a sample, and is assessed using the probability matrix (p_gt and p_lt). In total, 21 species pairs were analyzed to determine the frequency of their co-occurrence across the samples and the nature of their association (+ve or -ve ).

Line 208. The title of Table 2 is replaced with Optimized PCR conditions for detecting pests and pathogens.

Line 211-212 replaced with Before attempting to analyze for the presence or absence of honey bee pathogens and pests, we optimized PCR for the amplification of various targets using either single or

Line 216 revised as (Supplementary Figure S2).  

Line 220 replaced 139 bp with 138 bp.

Line 247 is replaced with “Additionally, 25 samples (19%) tested negative for all pathogens and pests”

Line 248-250 “The most recent national survey of honey bee pathogens and pests, last conducted in 2014, reveals a significant shift in their levels and distribution across Australia over the following seven years.”

Line 252 replaced with bacterial, fungal pathogens, and arthropod pests throughout Australia over the two years .

Line 259 replaced with Regional variations in pest and pathogen prevalence were evident across different .

Line 299 (Title 3.3 ) replaced with Pest and Pathogen Prevalence on Kangaroo Island.

Line 303 replaced with bacterial, fungal pathogens, and arthropod pests.

Line 311-316 replaced with The analysis revealed that 30% (40/135) of the samples contained a single type of pest or pathogen, while 20% (27/135) displayed two distinct types. Co-infections involving 3 and 4 different pests or pathogens were identified in 13% (18/135) and 10% (13/135) of the samples, respectively. Furthermore, 6% (8/135) contained 5 different types, with less than 4% (6/135) showing 6 distinct pest or pathogens. Notably, none of the samples tested positive for all seven types analyzed (Figure 4A).

Line 320 replaced with strong to moderate positive co-occurrence with all studied pests and pathogens

Line 334-336 replaced with Among the 21 pairs examined, including pest-pest, pathogen-pathogen, and mixed pest-pathogen combinations, 13 pairs exhibited positive associations, ranging from weak to strong, and tended to co-occur more frequently than expected.

Line 337-339 replaced with In contrast, eight pairs did not show statistically significant positive or negative association, suggesting no evidence of interaction between the involved pests, and/or pathogens.

Line 349-350 replaced with "that these organisms occur independently of each other, with no strong evidence of biological interaction. "

Line 351 replaced with "Figure 4. Distribution and co-occurrence of pests and pathogens in a honey sample. "

  1. Aethina tumida, and Galleria mellonella are not arthropod parasites, they are pests - a parasite is varroa.

Thank you very much for pointing this out. We acknowledge the error in referring to Aethina tumida and Galleria mellonella as arthropod parasites. As correctly noted, they are pests rather than parasites. The revised manuscript has corrected this to accurately reflect their classification, and we appreciate your attention to this important distinction.

  1. Nosema apis and Nosema ceranae are microsporidians, unicellular fungal endoparasites and so on. All definitions of the investigated biological threats should be more precise. 

Thank you so much for the feedback. Changes have been amended in all the passages that mention Nosema spp., as fungi, into microsporidian fungi

  1. Introduction: line 78: Conventional PCR is end-point PCR.

Thank you so much for the feedbacb comment. Line 83 replaced conventional PCR with end-point PCR. The text now reads:

“End-point PCR is the most widely used method for detecting bacteria (Paenibacillus larvae, Melissococcus plutonius), fungi (Nosema spp., Ascosphaera apis), and arthropods (Aethina tumida and Galleria mellonella) from honey and other hive samples [1-4].”

  1. Introduction: lines 86-88. This sentence is in a loop: DNA in honey is first extensively mentioned here , and then it is repeated from lines 94 to 107.  Reading the introduction, it seems the same point is repeated twice and disrupt the flow of the presentation of the main objective of the study. Introduction should be adjusted to avoid this loop. 

Thank you for the feedback. The redundant statement has been removed, and lines 96 to 100 (previously 94 - 98) have been revised to eliminate content that was already mentioned earlier.The text now reads:

“Honey eDNA has thus emerged as a practical method for monitoring honey bee pathogens, pests, and parasites [32, 33]. The genetic material left by organisms in the environment, known as eDNA, is a persistent biomolecular marker that can be collected, extracted, and analyzed from various substrates, making it a powerful tool for detecting and monitoring both microbial and macrobial communities effectively”

  1. M&M: line 119; what is the meaning of the asterisk? - in one region, just one honey sample has been collected. The relevance of the information derived is very low in the context.

Thank you for your comment. The asterisk has been removed, and a clarifying sentence has been added to provide clear information regarding sample collection. Additional passage has been added to the materials and method an the text now read as:

“Line 122 -126 Only one sample was obtained from Northern Territory (2762 registered hives) due to limited shipping options to our lab. Kangaroo Island is famous for its Ligurian honey bee sanctuary. Established in 1885, it preserves the purest strain of Apis mellifera ligustica by preventing crossbreeding. Biosecurity laws prevent the introduction of other bees and bee products to maintain the genetic purity of the bee population, making it a unique, valuable site for pest and pathogen surveillance.”

  1. M&M: line 124: The reference Rathinasamy et al. is not appropriate - it does not refer to honey

Thank you so much for the feedback comment. Line 130 - reference Rathinasamy et al. is removed from the manuscript.

  1. M&M: lines 125-137. It seems that the pre-treatment description derives from another reference not cited here.

Thank you so much for the feedback.The pre-treatment phase is pre-treatment phase is adopted from “Soares, S.; Amaral, J.S.; Oliveira, M.B.P.; Mafra, I. Improving DNA isolation from honey for the botanical origin identification. Food Control, 2015,48,130-136.” This reference is now added at Line 133.

9) M&M: line 188: it is not clear what p_gt is

Thank you for your feedback.The text has been altered to make this definition clearer.The text now read as:

This model includes combinatorial methods that calculate the probability of the observed frequency of co-occurrence (p_gt ). A p_gt value significantly less than expected (p_gt < 0.05), indicates a positive association (+ve) and with p_gt > 0.05 indicating negative association (-ve), or independent co-occurrence (p_lt > 0.05), which indicates no evidence of species interaction and suggests no significant association.

10) Table 1 and line 174: it is not explained why A. mellifera was tested - this is only reported in Results.

Thank you so much for the feedback to add valid point regarding testing A. mellifera . The reason for testing A. mellifera in this study is to assess the success of environmental DNA (eDNA) extraction and purification from honey samples. Specifically, we performed end-point PCR amplification of A.mellifera mitochondrial DNA (mtDNA) as a validation step. This was essential to confirm the efficacy of the eDNA extraction method prior to analyzing the eDNA from other species. The table 1 thus provides primer information for A. mellifera .To clarify this in the manuscript, we have added the following line to the Materials and Methods section: “Successful eDNA extraction and purification was determined by amplification of Apis mellifera mtDNA by end-point PCR.”

11) Table 1: it is not clear what is "suppl" in the correspondence of A. mellifera. There is no accession no. for A. mellifera

Thank you so much for pointing out this topography error.

Gene region updated : The gene region information for Apis mellifera was previously listed incorrectly. Replaced the suppl with mtDNA COI-COII, which is the gene region information required in that section of the table.

Accession number added : The accession number EF033649.1 has now been included under the accession number for Apis mellifera .

Product Size Correction: In the table, we had previously listed the product size for Apis mellifera as 139 bp. However, the correct size is 138 bp and replaced 139 with 138 under the product size for A. mellifera.

12) Lines 209-214: a reference should be cited as this test has been developed by others.

Thank you very much for your thoughtful comment. As the references for each individual PCR assay targeting specific pests and pathogens are already clearly provided in Table 1, we have excluded them from the main text to maintain clarity and avoid repetition. However, we are happy to include them in the main body if it would enhance the manuscript.

13) lines 239-241: A reference is missed. 

Thank you very much for your comment. Line 239-241 (Now 259-260) Reference is added supporting the The most recent national survey of honey bee pathogens and pests, last conducted in 2014, reveals a significant shift in their levels and distribution across Australia over the following seven years

14) Figure 2: How is it possible to compare the results of a study in 2013-2015 with what reported here? This should be elaborated into the text and mentioned in M&M

Thank you so much for the feedback. A section is added to the materials & methods, and now reads as:

“2.5. Assessing changes in pests and pathogen prevalence over time

To assess changes in the pathogen prevalence over time, we used the data from a study conducted between 2013 - 2015 on 154 adult bees in Australia as a reference point [5]. While our study utilised honey-derived eDNA, we evaluated potential shifts in the prevalence and distribution of these biological threats across the two time periods, supported by a bar diagram comparing the only available prior data with our findings.

15) Tables 3, 4 and 5: Northern Territory: only 1 sample - disclaimers should be clearly reported in the results related to this and discussed.

Thank you for highlighting this important point. We have clarified the sample size limitation in the Materials and Methods (Section 2.1), and added Supplementary Figure 1—a map of Australia showing the geographic distribution of honey samples. The following justification has been included: “Only one sample was obtained from the Northern Territory (2762 registered hives) due to limited shipping options to our lab.” Additionally, this limitation is now acknowledged in the Discussion section, where we state: “The inclusion of only one sample from the Northern Territory limits the comprehensiveness of our surveillance and regional inference.”

16) Lines 271-272: what is variance in prevalence? a statistica test should be reported

We appreciate the reviewer’s suggestion. In this analysis, we compared prevalence rates descriptively based on the total number of samples tested in each state, the number of positive cases identified, and the calculated percentages. The observed rates in NSW (59%) and QLD (58%) were nearly identical, and visual comparisons of raw counts and percentages supported the conclusion that there was no apparent difference. Given the minimal difference and the descriptive nature of this comparison, a formal statistical test was not conducted.

Line  291-293 (Previously 271 and 272) is now replaced with “The prevalence rates in NSW (59%) and QLD (58%) were similar, suggesting no apparent difference based on the identification of positive cases and calculated percentage.”

17) Lines 247-256: acronyms for the various regions should be defined at first use and also reported in Tables 3, 4 and 5 to simplify the reading.

Thank you for the feedback. We have now defined all regional acronyms (e.g., VIC, NSW, QLD) at their first mention in the text and ensured consistent use across Tables 3, 4, and 5. To improve readability with footnote added in table 3 and mentioned as "State abbreviations as defined in Table 3 ".below table 4 and 5.

18) Lines 355: end-point PCR (?)

Thank you so much for the feedback. Line 376 (Previously 355) the text replaced as “ To achieve this, we expanded existing end-point PCR assays to detect seven key disease-causing organisms relevant to honey bee health.”

19) Lines 396-388: a reference is missed

Thank you for noting this oversight. We have now added the appropriate reference to support the statement, which is included in Lines 403–405 (previously 396–388) of the revised manuscript.

20) Lines 414: Chalkbrood agent was detected not Chalkbrood

Thank you for your observation. The sentence has been revised to accurately state “The chalkbrood agent” in Line 435 (previously Line 414) of the updated manuscript.

21) Lines 418-419: Not clear what is referred to, Is it another study? In that case, a reference should be included.

Thank you for this clarification. We have revised the sentence to improve clarity and have now included appropriate references to support the global distribution of chalkbrood. This reference is now added at Line  438

The text now read as: Globally, chalkbrood is distributed across major beekeeping regions, including Central America, North America, Mexico, Chile, Japan, China, Turkey, Africa, and the Philippines.

22) Line 429: these acronyms should be defined

Thank you for the suggestion. We have now defined the acronym in the line 478 American Foulbrood (AFB) and “WOAH” as the World Organization for Animal Health at their first mention in the revised manuscript.

23) Line 432-433: italics for P. larvae

 Thank you for pointing this out. We have corrected the formatting and italicized P. larvae 

24) Lines 442-443: this comment is not clear as it is not reported clearly the comparative analysis between studies.

Thank you for your comment. We have revised the sentence to clarify the comparative context. Line 466-468 (Previously 442-443).The updated text now reads: “These findings reflect a broader global trend, with diverse patterns of AFB and EFB prevalence reported across different regions. In Indonesia, P. larvae was not detected, but M. plutonius was identified for the first time from honey and worker bee samples using end-point PCR.”

25) Line 446: Again, the mentioned study reported the presence of the DNA of the bacteria causing the mentioned diseases. This should be corrected in other parts of the discussion

Thank you for your feedback comment We have revised the sentence in Line 446 now Line 471

The updated sentence now reads:Similarly, P. larvae, DNA and spore were detected in 40% of the analyzed honey samples in Italy by end-point PCR and spore count, indicating a long-term persistence of the disease in hive products

26) Lines 447-448: not clear the context and this results was obtained in the mentioned study. This should be clarified.

Thank you for highlighting this ambiguity. We have revised the sentence to clarify that the reported prevalence rates originate from the cited U.S. study [85]. Line 473- 475 (Previously 447-448 ), The updated sentence now reads: A study in the U.S reported that, EFB (19.2%) was more common than AFB (8.6%) in symptomatic colonies, despite the resilience and ease of spread of P. larvae spores.”

27) Lines 465-466: significant positive trend?

Thank you for your observation. We have clarified the text to specify that the comparison indicates an increasing trend in the prevalence of SHB and wax moth over time.

Line 498-500 (Previously 465-466) replaced and the text now read as: “A comparison with earlier Australian data (2013-2015), based on adult bee samples, reveals a current upward trend in both arthropod pests.”

28) Discussion: there is no mention to the fact that the co-occurence is detected on honey samples; this does not mean that the co-occurence is also at the colony level as honey can derived from more colonies and apiaries. This should be clearly mentioned.

Thank you for this important observation. We have revised the Discussion section to clarify that co-occurrence patterns were identified in pooled honey samples, which represent an apiary-level overview rather than colony-specific dynamics. The sentence now reads: As the honey samples obtained in this study were pooled from multiple hives, the results represents an apiary-level snapshot, rather than colony-specific data, limiting the detection of active infection or co-occurence patterns at the individual hive level” (Line 554-556)

29) Some limitations of the study should be reported: the low number of samples tested, the need to include additional pathogens or parasites (for example varroa) and the need to georeference the samples to have a more precise map of the infected regions.

Thank you for this insightful comment. We have expanded the Discussion to address these limitations.

The text now included in Line 514-518

“ While this study focused on the detection of SHB and wax moth, other significant arthropod pests that pose serious threats to honey bee health globally - Varroa destructorTropilaelaps clareae, and Acarapis woodi were not targeted. Their exclusion represents a limitation, especially given their global spread and potential introduction into Australian apiaries through trade and hive movement .”

Line 526-528: “ However, investigation into co-occurrence in bee hives is limited. Even when co-occurrence is detected in honey samples, it does not necessarily indicate that the same pests and pathogens coexist within individual colonies, as honey may be sourced from multiple hives.”

Line 554- 556 “As the honey samples obtained in this study were pooled from multiple hives, the results represents an apiary-level snapshot, rather than colony-specific data, limiting the detection of active infection or co-occurence patterns at the individual hive level.”

30) References. Another important study that analysed honeyeDNA to monitor the diffusion of a parasite is not mentioned and discussed. This study is relevant for the epidemiological information derived from honey eDNA: https://doi.org/10.1016/j.jip.2021.107628.

Thank you so much for the feedback. A small statement has been included mentioning parasites included in this study.The text read as : Furthermore, the exclusion of certain key arthropod pests and parasites, along with the inclusion of only one sample from the NT, further limit the comprehensiveness of our surveillance and regional inference.”

Reviewer 2 Report

Comments and Suggestions for Authors

Nationwide Screening for Arthropod, Fungal, and Bacterial Honey Bee Pathogens: Utilizing Environmental DNA from Honey Samples in Australia

The manuscript submitted to Insects reveals widespread presence of pathogens and parasites in Australian honey bee colonies addresses an important issue in honey bee health monitoring. The study explores the use of honey-derived eDNA for the detection of honey bee pathogens across Australian states. The authors provided firm explanation for using honey as sample, which is one of the keystones of the article.

The authors used conventional singleplex and multiplex PCR assays for several pathogens and pests. The manuscript is clearly written and presents results that support the use of honey as a valuable non-invasive tool for monitoring colony health and detecting invasive species. However, some methodological and conceptual issues need to be addressed before the manuscript is suitable for publication. The research topic is current with potential for practical application.

The paper can be considered for acceptance after major revision.

Below are my specific comments and questions about the manuscript:

The title: Is it correct to consider the arthropods as pathogens or rather as pests? Having in mind that in current literature they are not usually classified as pathogens, there is a need for providing further explanation early in the text on why are you considering the arthropods Aethina tumida and Galleria mellonella as pathogenic.

Lines 43–46: Please cite more recent studies supporting the use of honey for pathogen detection, including ones that used metagenomic or next-gen sequencing approaches.

Lines 72-72: In my opinion, the way the sentence is written can cause the confusion, so it should be rephrased.

Line 99: With this abbreviation previously being introduced in the paper, there is no need for further use of the full term.

Line 117: When explaining sampling and number of samples taken from different locations, the first thing to mention should be total number of obtained samples. That way, the total scope of the research is clear before proceeding to the locations individually.

Lines 204-208: In my opinion, this paragraph is more thematically suited to the “Materials and Methods” section, as the optimization step typically performed prior to conducting the PCR.

Line 217: Here, it is stated that “All samples except 104 tested positive for N. ceranae”, but then, in the line 235 says that N. ceranae was present in 57% of the samples. Having in mind that those data aren’t aligned, you must rephrase this result to avoid possible confusion.

Lines 226-227: If the abbreviations for pathogen species were already used before, there is no need for using the full terms further in the paper.

Table 3: The name of the third column should be written as „No. of Samples Positive for P.larvae” so that it is explained what the numbers below represent. It may be obvious but the form should be precise.

State: It may be beneficial to the paper if there was a scheme of Australia with marked geographical regions from which the samples were collected. That way, non-Australian readers can have a better perception and that may provide better understanding of the sample collection sites.

Table 3, 4 and 5: There is no need for noting beneath every table that “Although Kangaroo Island is part of South Australia, data is shown for KI due to its unique honey bee population’’. It can be explained in the text, when explaining areas from which honey samples originated or included in the separate paragraph about the Kangaroo Island.

Line 274: The data about the samples with mixed infection with both N.ceranae and N. apis can be also shown in the table, but as a separated column in the end.

Line 287: I'm not sure if this is correct place for this paragraph. It does provide the explanation on why Kangaroo Island is shown as separate area even though, geographically, it isn't. However, you may consider placing the paragraph somewhere earlier in the ''Material and Methods" section so it is more clear why this region is mentioned separately in the Tables 3-5.

Discussion: The discussion has a descriptive character, stating the prevalence of pathogens by region, but does not relate the results to the aim of the study. Much of the discussion reads as an extended summary of existing literature, particularly in the sections on Nosema ceranae, without a critical evaluation. Please reformulate the discussion so that you relate your results more closely to the current results in the literature.

Lines 370-371: Please, support with a reference.

Lines 396-402: Can you find a more recent reference for this finding?

Lines 414-415: Please rephrase. The presence of A. apis, whether and where it was detected, was confusingly stated. Be more specific in which region it was identified and in what percentage.

Conclusion: The conclusion lacks proposals to future research that would result from this study. I suggest 1-2 sentences about the limits of the study, as well as about the possibilities of expansion and conducting future research.

Author Response

The title: Is it correct to consider the arthropods as pathogens or rather as pests? Having in mind that in current literature they are not usually classified as pathogens, there is a need for providing further explanation early in the text on why are you considering the arthropods Aethina tumida and Galleria mellonella as pathogenic.

Thank you for your valuable feedback. We agree that arthropods such as Aethina tumida and Galleria mellonella are more appropriately described as pests rather than pathogens. To address this, we have updated the manuscript title to: “Nationwide Screening for Arthropod, Fungal, and Bacterial Honey Bee Pathogens and Pests: Utilizing Environmental DNA from Honey Samples in Australia.”

Additionally, we have clarified this distinction in the Introduction considering Aethina tumida and Galleria mellonella as pathogenic. The text now included in Line 83-85

“Although, arthropods are not true pathogenic, their significant pest burden impact honey bee colony health.”

Lines 43–46: Please cite more recent studies supporting the use of honey for pathogen detection, including ones that used metagenomic or next-gen sequencing approaches.

Thank you for your helpful suggestion. In response, we have revised the introduction (lines 43–46) to include recent studies that apply metagenomic and next-generation sequencing (NGS) approaches for pathogen detection using honey samples lines 103 -107.

Lines 72-72: In my opinion, the way the sentence is written can cause the confusion, so it should be rephrased.

Thank you for your feedback. Line 72: is replaced withAmerican foulbrood (Paenibacillus larvae) and European foulbrood (Melissococcus plutonius), which are brood diseases that compromise larval health and weaken overall colony vitality.

Line 99: With this abbreviation previously being introduced in the paper, there is no need for further use of the full term.

Thank you for your suggestion. The full term has been removed, and the abbreviation “eDNA” is now used consistently from Line 100 onward in the revised manuscript

Line 117: When explaining sampling and number of samples taken from different locations, the first thing to mention should be total number of obtained samples. That way, the total scope of the research is clear before proceeding to the locations individually.

Thank you for your suggestion. We have revised the sentence to clearly introduce the total sample number upfront. The updated sentence now reads:

A total of 135 honey samples were gathered between 2022 and 2024 from diverse regions across Australia, including trade markets and directly from beekeepers, with regional distribution and sampling location to ensure a comprehensive representation of honey sources (Supplementary Figure S1)

Lines 204-208: In my opinion, this paragraph is more thematically suited to the “Materials and Methods” section, as the optimization step typically performed prior to conducting the PCR.

Thank you for your feedback. The content from Lines 204–208 has been removed from the Results section and incorporated into the Materials and Methods under the “PCR Analysis” subsection. The updated text (Lines 180–184) now reads: “The sensitivity of each primer set was validated through serial dilution of synthetic plasmid DNA from 10⁻¹ ng/µL to 10⁻⁹ ng/µL concentration, with the lowest detectable limit being 10⁻⁵ ng/µL. Specificity was evaluated using genomic DNA of P. larvae, M. plutonius, A. apis, N. apis, N. ceranae, A. tumida, and G. mellonella, with no cross-reactivity observed.”

Line 217: Here, it is stated that “All samples except 104 tested positive for N. ceranae”, but then, in the line 235 says that N. ceranae was present in 57% of the samples. Having in mind that those data aren’t aligned, you must rephrase this result to avoid possible confusion.

Thank you for identifying this inconsistency. The author understands the confusion. we have revised the Line 234- 239 (Previously 217) The updated text now provides representative results for a subset of samples rather than implying full-sample prevalence. Specifically, we state: Representative examples of data from samples 104 to 110 are shown in Figure 1A and samples 125 to 131 in Figure 1B. Samples 107 and 108 were positive for N. apis, while samples 105–110 tested positive for N. ceranae (Figure 1A). In the multiplex PCR targeting bacterial pathogens P. larvae, M. plutonius, and fungal pathogen A. apis, sample 107 amplified all three pathogens, whereas sample 108 tested positive for P. larvae and M. plutonius.”

Lines 226-227: If the abbreviations for pathogen species were already used before, there is no need for using the full terms further in the paper.

Thank you for the suggestion. We have updated the manuscript accordingly by replacing the full terms with their respective abbreviations in Line 245- 248 (Previously 225-228) now read as PCR products amplified for the A. mellifera, P. larvae, M. plutonius, A. apis, N. apis, and N. ceranae from honey samples 104 to 110, with the corresponding size shown in brackets. B) PCR products amplified for the A. mellifera, A. tumida, and G. mellonella from honey samples 125 to 131, with the corresponding size shown in brackets.

Table 3: The name of the third column should be written as „No. of Samples Positive for P.larvae” so that it is explained what the numbers below represent. It may be obvious but the form should be precise.

Thank you for your helpful observation. We have revised the column heading in Table 3, Table 4, and Table 5, adding “Number of samples positive for”.

State: It may be beneficial to the paper if there was a scheme of Australia with marked geographical regions from which the samples were collected. That way, non-Australian readers can have a better perception and that may provide better understanding of the sample collection sites.

Thank you for this valuable suggestion. We have added a map of Australia as a supplementary figure, showing the number of hives present and the number of samples collected from each state to provide clearer geographical context (Supplementary Figure 1).

Table 3, 4 and 5: There is no need for noting beneath every table that “Although Kangaroo Island is part of South Australia, data is shown for KI due to its unique honey bee population’’. It can be explained in the text, when explaining areas from which honey samples originated or included in the separate paragraph about the Kangaroo Island.

Thank you for the helpful suggestion. We have removed the repetitive note beneath Tables 3, 4, and 5. Instead, the unique status of Kangaroo Island’s honey bee population is now clearly explained in the Materials and Methods section.The revised text in Materials and Methods section now read as : Kangaroo Island is famous for its Ligurian honey bee sanctuary. Established in 1885, it preserves the purest strain of Apis mellifera ligustica by preventing crossbreeding. Strict biosecurity laws prevent the introduction of other bees and bee products to maintain the genetic purity of the bee population, making it a unique, valuable site for pest and pathogen surveillance.  

Line 274: The data about the samples with mixed infection with both N.ceranae and N. apis can be also shown in the table, but as a separated column in the end.

Thank you for the suggestion. We have incorporated a separate column in Table 4 to present the number of samples with mixed infection of N. ceranae and N. apis, displayed state-wise along with the total at the end of the table for clarity.

Line 287: I'm not sure if this is correct place for this paragraph. It does provide the explanation on why Kangaroo Island is shown as separate area even though, geographically, it isn't. However, you may consider placing the paragraph somewhere earlier in the ''Material and Methods" section so it is more clear why this region is mentioned separately in the Tables 3-5.

Thank you for the thoughtful feedback. As suggested, we have repositioned the explanation about Kangaroo Island earlier in the Materials and Methods section (Lines 122–124) to clarify its unique status to justifies its separate representation in the results.

The revised text in Materials and Methods section now read as : Kangaroo Island is famous for its Ligurian honey bee sanctuary. Established in 1885, it preserves the purest strain of Apis mellifera ligustica by preventing crossbreeding. Strict biosecurity laws prevent the introduction of other bees and bee products to maintain the genetic purity of the bee population, making it a unique, valuable site for pest and pathogen surveillance.  

Discussion: The discussion has a descriptive character, stating the prevalence of pathogens by region, but does not relate the results to the aim of the study. Much of the discussion reads as an extended summary of existing literature, particularly in the sections on Nosema ceranae, without a critical evaluation. Please reformulate the discussion so that you relate your results more closely to the current results in the literature.

Lines 370-371: Please, support with a reference.

Thank you so much for the feedback. This findings derived directly from our study’s results and restated to avoid confusion.The updated sentence reads: In comparison, our current data highlight a shift in prevalence, with N. ceranae, A. tumida, and G. mellonella emerging as the most widely distributed and frequently detected pathogens and pests across Australia. Furthermore, the bacterium P. larvae, and M. plutonius, continue to detect at low prevalence, while the microsporidian N. apis, and fungus A. apis showed reduced prevalence compared to the previous data.

Lines 396-402: Can you find a more recent reference for this finding?

Thank you for your feedback. This section has been revised to incorporate more recent findings and references.The updated text Line 403-406 (Previously 396-402 ) now read as :Our findings show that N. ceranae has largely dominated over N. apis in Australia, reflecting a broader global trend [18, 58-68]. However, the overall prevalence of N. ceranae and N. apis detected in our honey samples was lower than the levels reported in adult bee samples from a previous Australian study conducted between 2013 – 2015 [2].

Lines 414-415: Please rephrase. The presence of A. apis, whether and where it was detected, was confusingly stated. Be more specific in which region it was identified and in what percentage.

Thank you for your helpful suggestion. To improve clarity, we have revised the sentence to specify both the detection rate and regional distribution of A. apis. The updated sentence (now Lines 436–440) reads: The chalkbrood agent, A. apis, was detected in 5% of the analyzed honey samples, primarily from VIC and NSW. Globally, chalkbrood is distributed across major beekeeping regions, including Central America, North America, Mexico, Chile, Japan, China, Turkey, Africa, and the Philippines [79, 80]. In Australia, A. apis was first reported in QLD and subsequently spread across all states [81].

Conclusion: The conclusion lacks proposals to future research that would result from this study. I suggest 1-2 sentences about the limits of the study, as well as about the possibilities of expansion and conducting future research.

Thank you for this helpful suggestion. We have revised the conclusion to include a concise summary of the study’s limitations and proposed directions for future research. The revised conclusion now reads as:

“For the first time, a detailed pathogen prevalence survey using honey from different states of Australia revealed that N. ceranae is the most prevalent pathogen, followed by A. tumida, G. mellonella, P. larvae, N. apis, M. plutonius, and A. apis. Detection of nosemosis on Kangaroo Island highlights emerging biosecurity threats. While honey provides a non-invasive method for monitoring, it limits interpretation of active infections or hive-level dynamics. More research is required to understand the co-occurence pattern observed in this study. Future research should expand on conducting routine surveillance every 3-5 years and further developing these assays into a rapid, field-deployable tool to support early detection at the hive level to enhance biosecurity responses.”

Round 2

Reviewer 1 Report

Comments and Suggestions for Authors

The manuscript has been improved in many parts. However, nmenclature is still not completely correct, there are some parts that needs to be refined. Additionally, the number of cied references is too much. there is a cetain redundancy in the citation and some important wors are not cited. The cited references should not be more than 90-95 in total. Supplementary material is very confusing and should be careully re-organised and corrected.

The following corrections should be included in the text:

1) Title: Arthropods are pests, therefore the order of some words should be changed to respect the correspondences between organisms and the type of threats:

"Nationwide Screening for Arthropod, Fungal, and Bacterial Honey Bee Pests and Pathogens: Utilizing Environmental DNA 3 from Honey Samples in Australia" 

2) Line 13: "...significant threats from bacteria, fungi, pathogens and arthropod pests...". Bacteria and fungi are pathogens, therefore the sentence should be modified.

3) Simple summary: it does not provide any results or relevant information: for instance, geographical coverage or some other information to better explain what was done. It should be substantially improved.

3a) A few information about the geographical distribution of the analysed honey samples is important, and it is not mentioned

4) Line 28: parasites? they could be considered the microsporiadians even if they are commonly refered to pathogens - please clarify this issue that would affect all other arts of the text, including title.

5) Line 31: Again: conventional PCR: it should be end-point PCR

6) Line 37: see comment related to the title

7) Keywords: only two pathogens are mentioned, others pathogens/pests are not included; there should be consistency

8) Introduction: lines 57-67: this should be the first part of the introduction and moved up - in the current position, it disrupt the logical flow of the information and does not make any sense.

9) Line 98: "Honey eDNA has thus emerged as a practical method..." actually, honey eDNA is not a method is a source of information - The cited reference here [32,33] are not the most appropriate.  [21] and similar papers are more appropriate, including https://doi.org/10.1016/j.jip.2021.107628. non yet cited in the previous revision, despite being the first example of monitoring at large scale some pathogens from honey eDNA

10) Line 114: parasites?

11) Line 154 - please check 17,000 x g - it seems too much for a bench centrifuge

12) Lines 224-232: again, this part needs to have a cited reference.

13) Figures 1 A: A. mellifera not A. Mellifera - N. ceranae not N. Ceranae

14) Line 255 - prevalent pathogen - however, A. tumida is a pest - the subseuent sentence does not hold

15) Lines 259-263: not clear from this text if sentences related to the shift refer to [2] or derives from the comparison between the previous study and the current study

16) Table 3 and lines 267-276 - it is still not clear the difference between the disease and the biological agents of the diseases - the comparison in the text should be clarified, Table 3 is still not clear in this context and should be refined.

17) Table 4: Mixed infection? not correct

18) Line 310 - ... for the three brood disease agents ...

19) Line 318: this is not a co-infection it is a co-occurence

20) Figure 4 A: Number of Pathogens ? only pathogens?

21) Line 423 strategies ... replacement

22) lines 432-433 ... but also a sampling bias (only 6 honey samples were analysed from KI

23) lines 455,456, 465 - what detected in this study were the agents of these diseases not the diseases - please revise here and in several other parts of the text - including lines 467,470,471 - check if at lines 475, 479, 486,487,488,538,539 the meaning of the sentences is correct

24) lines 496, 501, 504, 506, and so on the alternate use of the latin name and acronyms for pests and pathogens is confusing

25) lines 516, 517 - V. destructor, Tropilaelaps and A. woodi are considered parasites

26) lines 524,525 - here, parasites can be mentioned

27) line 568: this is not a detailed pathogen survey - it is an initial survey of the diffusion of a few pathogens and pests 

28) Supplementary materials: it is just a ppt available. However, the information reported should be organised in more meaningful way, in more than one file. or in just one word file that will be then transformed into a pdf file

  • The title of the article should be corrected also in the supplementary material
  • All information related to the sensitivity assay should be combined in just one figure page - legend should be substantially improved. A figure with the specificity assay should be combined in just one figure page (it is another supplementary figure) - what is NTC - all figures and legends should be self explanatory
  • There is still confusion of the terms pathogens and pests
  • It is not clear why some pathogens/pests are written with the complete name and others are abbreviated
  • Supplementary Figure 3: in just one figure page or organised in a different way: it is not clear what are the numbers at the top of each gels and if there is a relations with the samples, what is the lin between this information and other amplification results or the samples collected in various regions. This information should be clarified. It is not clear what is the meaning of "+ve".
  • Supplemenatry Figure 4. The same comments reported above - please improve and correct the legend (Edna?) - the santence is probably incomplete
  • Slides 23, 28, 31 - why "PCR for the detection of ..." only here ? - please re-organise the results of the gels and the figures.
  • A table summarizing the results obtained from all 135 samples with information from the positive and negative samples for each tested pathogen and pest should be reported
  • Supplemenary table S1: in words or excel file - only pathogen co-occurrence ? in the legend and table nomenclature - effects size and other columsn  reduce the number of decimal to be consistent among columns - the names of the Pathogen 1 and Pathogen 2 (Pathogen/Pest) should be corrected  - there are errors
  •  
  •  
Comments on the Quality of English Language

Some sentences are not cmpletely clear or incomplete. Please revise carefully the text

Author Response

The following corrections should be included in the text:

1) Title: Arthropods are pests, therefore the order of some words should be changed to respect the correspondences between organisms and the type of threats:

"Nationwide Screening for Arthropod, Fungal, and Bacterial Honey Bee Pests and Pathogens: Utilizing Environmental DNA 3 from Honey Samples in Australia" 

Thank you so much for the feedback. The words are now arranged to match the correspondace between the organism and the type of threat.

The title now read as “Nationwide Screening for Arthropod ,Fungal, and Bacterial Pests and Pathogens of Honey Bees: Utilizing Environmental DNA from Honey Samples in Australia”

2) Line 13: "...significant threats from bacteria, fungi, pathogens and arthropod pests...". Bacteria and fungi are pathogens, therefore the sentence should be modified.

Thank you so much for the feedback. The sentence has now been corrected for clarity and read as “Honey bees face significant threats from bacteria and fungal pathogens and arthropod pests”

3) Simple summary: it does not provide any results or relevant information: for instance, geographical coverage or some other information to better explain what was done. It should be substantially improved.

3a) A few information about the geographical distribution of the analysed honey samples is important, and it is not mentioned

Thank you so much for the feedback. In response, the simple summary has been improved by providing more detailed information on the sample size and stating samples were systematically collected from all states and territories across Australia. Additionally, we have included data on the prevalence rates of pests and pathogens,nationwide. The added passage read as

“For this study, a total of 135 honey samples were collected from across all states of Australia, representing a broad geographical distribution. The most prevalent pathogen detected was Nosema ceranae (57%), followed by the pests Aethina tumida (40%) and Galleria mellonella (37%), and the pathogens Paenibacillus larvae (21%), Nosema apis ( 19%) and Melissococcus plutonius (18%). The prevalence of Ascosphaera apis was low (5%) across Australia. These findings provide essential information on the regional distribution of key pests and pathogens across Australia.”

4) Line 28: parasites? they could be considered the microsporiadians even if they are commonly refered to pathogens - please clarify this issue that would affect all other arts of the text, including title.

Thank you so much for the feedback. We have revised the sentence to provide more specificity “ In this study, we extracted eDNA from 135 honey samples and tested for the presence of DNA for seven key honey bee pathogens and pests”.

To maintain consistency throughout the manuscript, we have chosen to refer to Nosema apis and Nosema ceranae as pathogens, while acknowledging their biological classification as microsporidian parasites. These organisms are unicellular fungi that are obligate intracellular parasites; however, given their established role in honey bee disease, the term “pathogen” remains widely used in the literature.

5) Line 31: Again: conventional PCR: it should be end-point PCR

Thank you so much for pointing this out. This has been now changed to End-point PCR and the line read as “using end-point singleplex and multiplex PCR assays. ”

6) Line 37: see comment related to the title.

Thank you so much for the feedback.We agree the importance of aligning each organism with its corresponding threat category.In this section, our intention was to highlight the key findings of the study by presenting the organisms in descending order of prevalence, starting with the most frequently detected disease-causing organism across all honey samples. This approach aims to emphasize the relative impact of each pathogen and pest based on how commonly they were identified, rather than grouping them strictly by organism to the type of threats. This ordering more effectively communicates the significance of each organism’s detection in the context of nationwide monitoring.

7) Keywords: only two pathogens are mentioned, others pathogens/pests are not included; there should be consistency.

Thank you so much for the comment. The keywords have been revised including all the pathogens and pests. It is now read as

“Keywords: Apis mellifera; eDNA; Honey; Paenibacillus larvae; Melissococcus plutonius; Nosema apis; Nosema ceranae; Ascosphaera apis; Aethina tumida; Galleria mellonella; health; surveillance”

8) Introduction: lines 57-67: this should be the first part of the introduction and moved up - in the current position, it disrupt the logical flow of the information and does not make any sense.

Thank you so much for the feedback. We agree that the paragraph in lines 57–67 fits more appropriately as the opening of the Introduction. To improve the logical flow and readability, we have moved this content to the beginning of the Introduction section. The revised Introduction now begins with this paragraph

“In Australia, the agricultural sector heavily relies on insect pollination, particularly from the western honey bees, Apis mellifera [1] which is essential for pollinating a diverse array of crops and enhancing agricultural productivity and food security nationwide [2]. The benefits of crop pollination extend beyond agriculture, exerting far-reaching impacts on the broader Australian community by facilitating pollination and positively influencing crop outcomes through their synergistic relationship with flowering plants [3]. Previous estimations indicate that the total value of paid and unpaid pollination services amounts to approximately AUD 1.2 billion annually in Australia [4]. The reliance on honeybees for optimal pollination extends to approximately two-thirds of horticultural crops in the country [5], particularly enhancing productivity and post-harvest storage qualities in fruits like apples, raspberries, and peaches [3, 6, 7].

    Australia is renowned for having one of the healthiest honey bee populations globally. ”

9) Line 98: "Honey eDNA has thus emerged as a practical method..." actually, honey eDNA is not a method is a source of information - The cited reference here [32,33] are not the most appropriate.  [21] and similar papers are more appropriate, including https://doi.org/10.1016/j.jip.2021.107628. non yet cited in the previous revision, despite being the first example of monitoring at large scale some pathogens from honey eDNA

Thank you so much for this valuable feedback. We acknowledge that referring to honey eDNA as a “method” was imprecise. In response we have revised the sentence to:

“Honey eDNA has thus emerged as a valuable resource for monitoring honey bee pathogens, pests, and parasites ”.

Additionally,we have updated the references to reflect the more appropriate citations suggested by the reviewer.

10) Line 114: parasites?

Thank you so much for the comment. We have revised the sentence in Line 114 , term “parasite” has been removed, and the sentence now reads:

“These samples were utilized to isolate eDNA using the bead-beating-silica DNA extraction method and identified pests and pathogens known to impact honey bee populations in Australia. ”

11) Line 154 - please check 17,000 x g - it seems too much for a bench centrifuge

Thank you so much for bringing this to our attention. We confirm that the value of 17,000 × g (which is equivalent to 13,000 RPM) is accurate for our protocol .The centrifugation step was performed using the Eppendorf Centrifuge 5425 R (Eppendorf Centrifuge 5425 R, Refrigerated, 24-Place, 6 Rotors, Max Speed 21,300 x g, For Samples from 0.2-5mL, Membrane Keyboard), a high-performance refrigerated benchtop centrifuge with a maximum centrifugal force of 21,300× g, as specified by the manufacturer.

12) Lines 224-232: again, this part needs to have a cited reference.

Thank you so much for pointing this out. While the relevant references were originally cited in Table 2, we agree that reiterating them in the main text improves clarity and readability. Accordingly, the citations have now been added directly within Lines 266–274 (Previously Line:224-232).

13) Figures 1 A: A. mellifera not A. Mellifera - N. ceranae not N. Ceranae

Thank you so much for pointing this typographic error. The Figure 1 A has been replaced with correct formatting and no errors in species names

14) Line 255 - prevalent pathogen - however, A. tumida is a pest - the subseuent sentence does not hold

Thank you so much for the comment. The sentence has been revised to distinguish clearly between pathogens and pests. It now reads: “Analysis of 135 honey samples collected from various locations in Australia revealed N. ceranae emerged as the most prevalent pathogen, present in 57% of the samples. This was followed by the pests A. tumida (40%) and G. mellonella (37%), and the pathogens P. larvae (21%), N. apis (19%), and M. plutonius (18%). ”

15) Lines 259-263: not clear from this text if sentences related to the shift refer to [2] or derives from the comparison between the previous study and the current study.

Thank you for the helpful feedback. We agree that the original phrasing may have caused ambiguity regarding the reference point for the observed shift.Since the shift derives from a comparison between the previous study and current sudy,we have revised the sentence to link the observed changes and the sentence now reads:

“A comparison between the current data and the most recent national survey of honey bee pathogens and pests, conducted in 2014 [9], reveals a significant shift in their levels and distribution across Australia over the subsequent seven years . ”

16) Table 3 and lines 267-276 - it is still not clear the difference between the disease and the biological agents of the diseases - the comparison in the text should be clarified, Table 3 is still not clear in this context and should be refined.

Thank you for your thoughtful comment. We acknowledge that the previous version of the text and Table 3 may not have clearly differentiated between the diseases (American and European foulbrood) and their causative biological agents (P. larvae and M. plutonius). To address this, we have refined both the text and the table caption .The revised text now reads:

“Regional variations in pest and pathogen prevalence were evident across different Australian states. Specifically, American foulbrood (AFB), caused by P. larvae and European foulbrood (EFB), caused by M. plutonius, were identified in Victoria (VIC), New South Wales (NSW), Queensland (QLD), and Tasmania (TAS) (Table 3). Notably, VIC exhibited a high prevalence, with 33% of the sample testing positive for P. larvae and 37% for M. plutonius. In NSW and QLD, there was no significant difference in the prevalence of P. larvae  (28% and 21%, respectively) and M. plutonius (14% and 13%, respectively). TAS had a prevalence of 25% for both P. larvae  and M. plutonius.  In contrast, Western Australia (WA) only harbours P. larvae (14%) and is free from M. plutonius, while South Australia (SA) exclusively hosts M. plutonius (29%), with P. larvae  absent. Similarly, M. plutonius was absent in WA and Kangaroo Island (KI) (Table 3).

Table 3. Distribution of bacterial pathogens P. larvae (Causative agent of AFB) and M. plutonius (Causative agent of EFB) across different states in Australia.”

17) Table 4: Mixed infection? not correct

Thank you for this valuable feedback. We appreciate the reviewer’s attention to the clarity of Table 4, particularly regarding the terminology used for dual pathogen detection. To address this concern, the term “mixed infection” has been replaced with “co-occurence” to more accurately describe samples in which both N. apis and N. ceranae were detected concurrently within the same honey sample. This change has been applied to the manuscript and the table.Additionally, lines 348–349 (previously lines 296–297) have been revised to align with the updated terminology and now read:

“Additionally, 16% of the samples tested positive for both N. ceranae and N. apis, indicating co-occurrence (Table 4).”

18) Line 310 - ... for the three brood disease agents ...

Thank you so much for the comment. In response, we have revised the sentence to clearly specify the causative agents of brood diseases. The updated sentence now reads:“The results were significant, as all samples tested negative for the three brood disease agents (P. larvae, M. plutonius, and A. apis), as well as the pests A. tumida and G. mellonella.”

19) Line 318: this is not a co-infection it is a co-occurence

Thank you so much for the comment. We agree that the term co-occurrence more accurately describes the simultaneous detection of multiple pests and pathogens within a single sample, without implying biological interaction or active infection. We have revised the title and now it reads as

“Trends in Co-Occurrence of Pathogens and Pests

This study uncovered a diverse array of honey bee pathogens and pests present in individual honey samples, highlighting a significant degree of co-occurrence across the surveyed samples. The analysis revealed that 30% (40/135) of the samples contained a single type of pest or pathogen, while 20% (27/135) displayed two distinct types. Co-occurrence involving 3 and 4 different pests or pathogens were identified in 13% (18/135) and 10% (13/135) of the samples, respectively. Furthermore, 6% (8/135) contained 5 different types, and fewer than 4% (6/135) showing 6 distinct pest or pathogens. Notably, none of the samples tested positive for all seven types analyzed (Figure 3A).”

20) Figure 4 A: Number of Pathogens ? only pathogens?

Thank you so much for the feedback comment. Figure 3A has now been updated (Figure number was incorrect, which is now revised in the manuscript- Replace number: 4 with 3), we have revised the axis titles and the colour legend title to more accurately reflect the data presented.

21) Line 423 strategies ... replacement

Thank you so much for pointing this typographic error. The term “replacemnet” has been corrected to “replacement”

22) lines 432-433 ... but also a sampling bias (only 6 honey samples were analysed from KI

Thank you so much for the feedback. We agree that limited number of samples from Kangaroo island introduces potential sampling bias. We have revised the statement and the text now read as:

“This likely reflects a combination of strong biosecurity enforcement and the limited pests and pathogen diversity observed in the small number of samples collected from KI. While only 6 honey samples were analysed from this region – introducing potential sampling bias, these findings, along with continued surveillance and legislative support, underscore the role of ongoing monitoring in preserving the health and resilience of the unique honey bee population on KI.”

23) lines 455,456, 465 - what detected in this study were the agents of these diseases not the diseases - please revise here and in several other parts of the text - including lines 467,470,471 - check if at lines 475, 479, 486,487,488,538,539 the meaning of the sentences is correct

Thank you for this important feedback. We acknowledge that our study focused on the detection of disease-causing agents (e.g., P. larvae and M. plutonius), rather than clinical diagnoses of American foulbrood (AFB) or European foulbrood (EFB). In response to this comment, we have revised the relevant sections of the manuscript to reflect this distinction more accurately. Specifically, the original references to AFB and EFB in lines- 267-278,  455–471,475, 479, 486–488, 538, and 539 (now corresponding to lines 313–324,  458, 537–539, 548, 550–551, 553–554, 558, 563, 587–588, 590, and 593 in the revised manuscript) have been updated to refer to the detection of P. larvae and M. plutonius, where appropriate.

24) lines 496, 501, 504, 506, and so on the alternate use of the latin name and acronyms for pests and pathogens is confusing

Thank you so much for the feedback. We have revised the lines to address the alternating use of Latin names and common names for arthropod pests. we now introduce each pest by its latin and common name upon first mention —Aethina tumida (the small hive beetle, SHB) and Galleria mellonella (the wax moth). Subsequent references use only the common name acronym to maintain readability. These changes have been applied consistently throughout the relevant sections, including lines- 596-597, 602,605,607 (Previously 496, 501, and 504–506).

25) lines 516, 517 - V. destructor, Tropilaelaps and A. woodi are considered parasites

Thank you so much for this helpful clarification. We agree that Varroa destructor, Tropilaelaps clareae, and Acarapis woodi are parasitic mites rather than general arthropod pests. The relevant sentence has been revised and now refers to them as “parasitic mites that pose serious threats to honey bee health globally.”

26) lines 524,525 - here, parasites can be mentioned

Thank you so much for the suggestion. In response, we have revised the sentence to include parasites alongside pest and pathogens. The test now read as :

“Since hive health is influenced by multiple factors, understanding co-occurrence between different pests, pathogens and parasites is critical, as such, interactions often contribute to colony collapse ”

27) line 568: this is not a detailed pathogen survey - it is an initial survey of the diffusion of a few pathogens and pests In response, we have revised the text and now read as :

Thank you so much for the feedback. We agree that the study represents an initial investigation into the distribution of pathogens and pests, rather than a comprehensive pathogen prevalence survey.The statemnet has now been revised and read as:

“For the first time, an initial pest and pathogen prevalence survey using honey samples collected from different states of Australia revealed that N. ceranae is the most prevalent pathogen, followed by the pests A. tumida and G. mellonella, and the pathogens P. larvae, N. apis, M. plutonius, and A. apis. ”

28) Supplementary materials: it is just a ppt available. However, the information reported should be organised in more meaningful way, in more than one file. or in just one word file that will be then transformed into a pdf file

  • The title of the article should be corrected also in the supplementary material

Title page has been removed from new supplementary data file

  • All information related to the sensitivity assay should be combined in just one figure page - legend should be substantially improved. A figure with the specificity assay should be combined in just one figure page (it is another supplementary figure) - what is NTC - all figures and legends should be self explanatory

Sensitivity and specificity gels were organised into single figures and placed in new supplementary data file with updated figure legands.

  • It is not clear why some pathogens/pests are written with the complete name and others are abbreviated.

This has been made consisted through out the figures.

  • Supplementary Figure 3: in just one figure page or organised in a different way: it is not clear what are the numbers at the top of each gels and if there is a relations with the samples, what is the lin between this information and other amplification results or the samples collected in various regions. This information should be clarified. It is not clear what is the meaning of "+ve".

Supplementary Figure 3 and 4 have been removed as they are raw images and can be now found at Figshare site. Raw images for gels can be found at hhtp:/doi.org/ 10.26181/29487338.

Statement has been added to Data Availability Statement

  • Supplemenatry Figure 4. The same comments reported above - please improve and correct the legend (Edna?) - the santence is probably incomplete

Supplementary Figure 3 and 4 have been removed as they are raw images and can be now found at Figshare site. Raw images for gels can be found at hhtp:/doi.org/ 10.26181/29487338.

Statement has been added to Data Availability Statement

  • Slides 23, 28, 31 - why "PCR for the detection of ..." only here ? - please re-organise the results of the gels and the figures.

Supplementary Figure 3 and 4 have been removed as they are raw images and can be now found at Figshare site. Raw images for gels can be found at hhtp:/doi.org/ 10.26181/29487338.

  • A table summarizing the results obtained from all 135 samples with information from the positive and negative samples for each tested pathogen and pest should be reported

Table summarizing the 135 samples has been included as Supplemtary Table 1 in revised manuscript.

  • Supplemenary table S1: in words or excel file - only pathogen co-occurrence ? in the legend and table nomenclature - effects size and other columsn  reduce the number of decimal to be consistent among columns - the names of the Pathogen 1 and Pathogen 2 (Pathogen/Pest) should be corrected  - there are errors

Table has been corrected

Round 3

Reviewer 1 Report

Comments and Suggestions for Authors

The manuscript has been substantially improved according to the indications.

There are still some typos that should be eliminated by reading twice the manuscript.

Additionally, in figure 1 - the gel showing the amplified bands for A. mellifera is not clear. The size of the amplified bands if related to the ladder (that was copied and pasted for all gels) does not provide accurate information of what has been indicated: 85 and 138 bp - it seems 152 and 138 bp instead - plese revise the position of the ladder.

Suppl. figure 2: M. Plutonius - M. plutonius

What is the red band in Suppl. Fig. 3?

Suppl. Table 1: the last row does not have any information: it is probably the sum of positive samples (?)

Suppl. Table 2. Is it a pdf copied in the word ? Why? it should be a Words table.

Author Response

Q1. There are still some typos that should be eliminated by reading twice the manuscript.

The manuscript has been reread, and several typos have been removed

Q2. Additionally, in figure 1 - the gel showing the amplified bands for A. mellifera is not clear. The size of the amplified bands if related to the ladder (that was copied and pasted for all gels) does not provide accurate information of what has been indicated: 85 and 138 bp - it seems 152 and 138 bp instead - plese revise the position of the ladder.

Figure 1 has been corrected to show each ladder band for each gel slice.

Q3. Suppl. figure 2: M. Plutonius - M. plutonius

  1. Plutonius has been corrected to M. plutonius

Q4. What is the red band in Suppl. Fig. 3?

The red bands are due to overexposure of the setting on the camera. This is done to allow for the detection of any faint bands.

Q5. Suppl. Table 1: the last row does not have any information: it is probably the sum of positive samples (?)

Sorry, the last row was missing a header. It has now been corrected to Total positive samples. 

Q6. Suppl. Table 2. Is it a pdf copied in the word ? Why? it should be a Words table.

The table has been changed from a PDF to a Word table.